# MiraData: A Large-Scale Video Dataset with Long Durations and Structured Captions

Xuan Ju[1,2*], Yiming Gao[1*], Zhaoyang Zhang[1†*], Ziyang Yuan[1], Xintao Wang[1],
Ailing Zeng[2], Yu Xiong[2], Qiang Xu[2], Ying Shan[1]
https://github.com/mira-space/MiraData

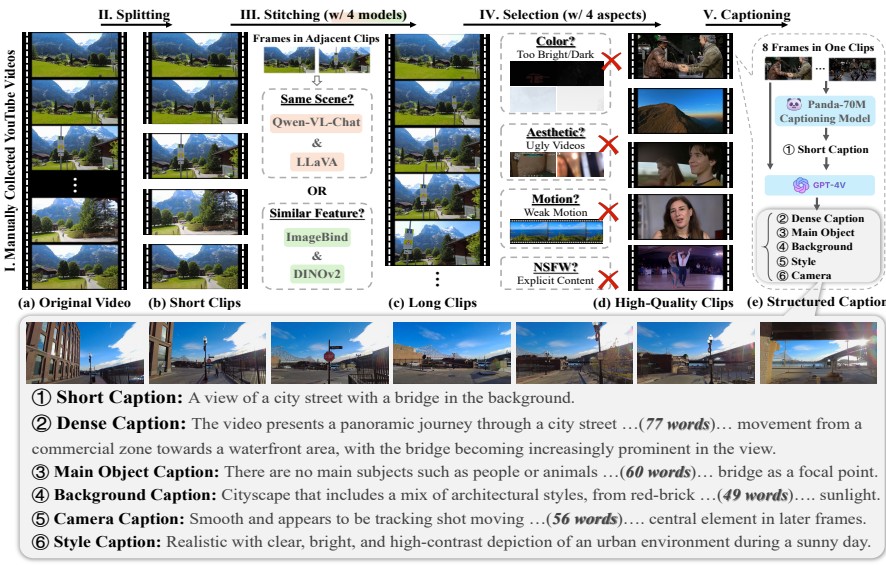

① **Short Caption:** A view of a city street with a bridge in the background.
② **Dense Caption:** The video presents a panoramic journey through a city street …(**77 words**)… movement from a commercial zone towards a waterfront area, with the bridge becoming increasingly prominent in the view.
③ **Main Object Caption:** There are no main subjects such as people or animals …(**60 words**)… bridge as a focal point.
④ **Background Caption:** Cityscape that includes a mix of architectural styles, from red-brick …(**49 words**)…. sunlight.
⑤ **Camera Caption:** Smooth and appears to be tracking shot moving …(**56 words**)…. central element in later frames.
⑥ **Style Caption:** Realistic with clear, bright, and high-contrast depiction of an urban environment during a sunny day.

Figure 1: **Video collection and annotation pipeline.** An example shown at bottom.

## Abstract

Sora's high-motion intensity and long consistent videos have significantly impacted the field of video generation, attracting unprecedented attention. However, existing publicly available datasets are inadequate for generating Sora-like videos, as they mainly contain short videos with low motion intensity and brief captions. To address these issues, we propose *MiraData*, a high-quality video dataset that surpasses previous ones in video duration, caption detail, motion strength, and visual quality. We curate *MiraData* from diverse, manually selected sources and meticulously process the data to obtain semantically consistent clips. GPT-4V is employed to annotate structured captions, providing detailed descriptions from four different perspectives along with a summarized dense caption. To better assess temporal consistency and motion intensity in video generation, we introduce *MiraBench*, which enhances existing benchmarks by adding 3D consistency and tracking-based motion strength metrics. *MiraBench* includes 150 evaluation prompts and 17 metrics covering temporal consistency, motion strength, 3D consistency, visual quality, text-video alignment, and distribution similarity. To demonstrate the utility and effectiveness of *MiraData*, we conduct experiments using our DiT-based video generation model, *MiraDiT*. The experimental results on *MiraBench* demonstrate the superiority of *MiraData*, especially in motion strength.

---

*Equal contribution. † Project Lead. [1]ARC Lab, Tencent PCG. [2]The Chinese University of Hong Kong.

Submitted to the 38th Conference on Neural Information Processing Systems (NeurIPS 2024) Track on Datasets and Benchmarks. Do not distribute.

# 1 Introduction

Recent advances in the Artificial Intelligence and Generative Content (AIGC) field, such as video generation [1, 2, 3], image generation [4, 5, 6, 7], and natural language processing [8, 9], have been rapidly progressing, thanks to the improvements in data scale and computational power. Previous studies [4, 9, 2, 7] have emphasized that data plays a pivotal role in determining the upper-bound performance of a task. A notable recent development is the introduction of Sora [1], a text-to-video generation model, shows stunning video generation capabilities far surpassing existing state-of-the-art methods. Sora not only excels in generating high-quality long videos (10-60 seconds) but also stands out in terms of motion strength, 3D consistency, adherence to real-world physics rules, and accurate interpretation of prompts, paving the way for even more sophisticated generative models in the future.

The first step in constructing Sora-like video generation models is the construction of a well-curated, high-quality dataset, as data forms the very foundation of model performance and capability. However, existing publicly video datasets, such as WebVid-10M [10], Panda-70M [11], and HD-VILA-100M [12], fall short of these requirements. These datasets primarily consist of short video clips (5-18 seconds) sourced from unfiltered videos from the internet, which leads to a large proportion of low-quality or low-motion clips and are inadequate for training generating Sora-like models. Moreover, the captions in existing datasets are often short (12-30 words) and lack the necessary details to describe the entire videos. These limitations hinder the use of existing datasets for generating long videos with accurate interpretation of prompts. Therefore, there is an urgent need for a comprehensive, high-quality video dataset with long video durations, strong motion strength, and detailed captions.

To tackle these issues, we present *MiraData*, a large-scale, high-quality video dataset specifically designed to meet the demands of long-duration high-quality video generation, featuring long videos (average of 72.1 seconds) with high motion intensity and detailed structured captions (average of 318 words). The data curation pipeline is illustrated in Fig. 1, where we have built an end-to-end pipeline for data downloading, segmentation, filtering, and annotation. **I. Downloading.** To obtain diverse videos, we collect source videos from manually selected channels of various platforms. **II & III. Segmentation.** We employ multiple models to compare semantic and visual feature information, segmenting videos into long clips with strong semantic consistency by using a mixture of models to detect clips within a video and cut long videos into smaller segments. **IV. Filtering.** To accommodate high-quality clips, we filter the dataset into five subsets based on aesthetics, motion intensity, and color to select clips with high visual quality and strong motion intensity. **V. Annotation.** To obtain detailed and accurate descriptions, we first use the state-of-the-art captioner [11] to generate a short caption and then employ GPT-4V to enrich it, resulting in the dense caption. To provide fine-grained video descriptions across multiple perspectives, we further design structured captions, which include descriptions of the video's main subject, background, camera motion, and style. To this end, statistical results encompassing video duration, caption length and elaboration, motion strength, and video quality demonstrate *MiraData*'s superiority over previous datasets.

To further analyze the performance gap between generated videos and high-quality real-world videos, we identify a crucial limitation in existing benchmarks: the lack of a comprehensive evaluation of 3D consistency and motion intensity in generated videos. To address this issue, we propose *MiraBench*, an enhanced benchmark that builds upon existing benchmarks by adding 3D consistency and tracking-based motion strength metrics. Specifically, MiraBench includes 17 metrics that comprehensively cover various aspects of video generation, such as temporal consistency, motion strength, 3D consistency, visual quality, text-video alignment, and distribution similarity. To evaluate the effectiveness of captions, we introduce 150 evaluation prompts in MiraBench, consisting of short captions, dense captions, and structured captions. These prompts provide a diverse set of challenges for assessing the performance of text-to-video generation models. To validate the effectiveness of our *MiraData* , we conduct experiments using our DiT-based video generation model, *MiraDiT*. Experimental results show the superiority of our model trained on *MiraData*, when compared to the same model trained on WebVid-10M and other state-of-art open-source methods on motion strength, 3D consistency and other metrics in *MiraBench*.

## 2 Related Work

### 2.1 Video-Text Datasets

Large-scale training on image-text pairs [13, 14, 15, 16, 17] has been proven effective in text-to-image generation [18, 19, 20] and vision-language representation learning [21, 22], showing emergent ability with model and data scaling-up. Recent achievements such as Sora [1] suggest that similar capabilities can be observed in the realm of videos, where data availability and computational resources emerge as crucial factors. However, previous text-video datasets, as shown in Tab. 1, are constrained by short durations, limited caption lengths, and poor visual quality.

Considering the domain of general video generation, a significant portion of open-source text-video datasets is unsuitable due to issues such as noisy text labels, low resolution, and limited domain coverage. Thus the majority of video generation models with impressive performance [23, 3, 24, 25, 26, 27, 28] rely heavily on internal datasets for training, which restricts transparency and usability. The commonly used open-source text-video dataset for video generation [29, 30, 31, 32, 33, 34, 35, 36, 37, 38, 39] is WebVid-10M [10]. However, it contains a prominent watermark on videos, requiring additional fine-tuning on image datasets (e.g., Laion [40]) or internal high-quality video datasets to remove the watermark. Recently, Panda-70M [11], InternVid [41], and HD-VG-130M [42] have been proposed and targeted for video generation. Panda-70M and InternVid aim to extract precise textual annotations using multiple caption models, while HD-VG-130M emphasizes the selection of high-quality videos. But none of them systematically considers correct video splitting, visual quality filtering, and accurate textual annotation at all three levels during the data collection process. More importantly, all previous datasets consist of videos with short durations and limited text lengths, which restricts their suitability for long video generation with fine-grained textual control.

Table 1: **Comparison of *MiraData* and pervious large-scale video-text datasets.** Datasets are sorted based on average text length. Datasets with gray background are used in a text-to-video generation. *MiraData* significantly surpasses previous datasets in average text and video length.

| Dataset | Avg text len | Avg / Total video len | | Year | Text | Domain | Resolution |
|---|---|---|---|---|---|---|---|
| HowTo100M [43] | 4.0 words | 3.6s | 135Khr | 2019 | ASR | Open | 240p |
| LSMDC [44] | 7.0 words | 4.8s | 158h | 2015 | Manual | Movie | 1080p |
| DiDeMo [45] | 8.0 words | 6.9s | 87h | 2017 | Manual | Flickr | - |
| YouCook2 [46] | 8.8 words | 19.6s | 176h | 2018 | Manual | Cooking | - |
| MSR-VTT [47] | 9.3 words | 15.0s | 40h | 2016 | Manual | Open | 240p |
| HD-VG-130M [42] | ∼9.6 words | ∼5.1s | ∼184Khr | 2024 | Generated | Open | 720p |
| WebVid-10M [10] | 12.0 words | 18.0s | 52Kh | 2021 | Alt-Text | Open | 360p |
| Panda-70M [11] | 13.2 words | 8.5s | 167Khr | 2024 | Generated | Open | 720p |
| ActivityNet [48] | 13.5 words | 36.0s | 849h | 2017 | Manual | Action | - |
| VATEX [49] | 15.2 words | ∼10s | ∼115h | 2019 | Manual | Open | - |
| HD-VILA-100M [12] | 17.6 words | 11.7s | 760.3Khr | 2022 | ASR | Open | 720p |
| How2 [50] | 20.0 words | 5.8s | 308h | 2018 | Manual | Instruct | - |
| InternVid [41] | 32.5 words | 13.4s | 371.5Khr | 2023 | Generated | Open | 720p |
| *MiraData* (Ours) | 318.0 words | 72.1s | 16Khr | 2024 | Generated | Open | 720p |

### 2.2 Video Generation

Video generation is a challenging task that have advanced from early GAN-based models [51, 52] to more recent diffusion. Diffusion-based methods have made significant progress in terms of visual quality and diversity in generated videos while entailing a substantial computational cost [24, 3]. Consequently, researchers often face a trade-off between the quality of the generated videos and the duration of the videos that can be produced within practical computational constraints.

To ensure visual quality under computational resource constraints, previous diffusion-based video generation methods primarily focus on open-domain text-to-video generation with a **short duration**. Video Diffusion Models [25] is the first to employ the diffusion model for video generation. To

generate long videos in the absence of corresponding dataset, Make-A-Video [29] and NUWA-XL [53] explore coarse-to-fine video generation but suffer from maintaining temporal continuity and producing strong motion magnitude. Apart from these explorations of convolution-based architecture [29, 30, 31, 25, 23, 27, 24, 32, 42, 37, 34, 35, 33, 38, 39], transformer-based methods (*e.g.*, WALT [26], Latte [54], and Snap Video [3]) become more prevalent recently, offering a better trade-off between computational complexity and performance, as well as improved scalability.

All previous methods can only generate short video clips (*e.g.*, 2 seconds, 16 frames) with weak motion strength. However, the recent success of Sora [1] demonstrates the potential of long video generation with enhanced motion strength and strong 3D consistency. With the belief that data is the key to machine learning, we find that existing datasets' (1) short duration, (2) weak motion strength, and (3) short and inaccurate captions are insufficient for Sora-like video generation model training (as shown in Tab. 1). To address these limitations and facilitate the development of advanced video generation models, we introduce *MiraData*, the first large-scale video dataset specifically designed for long video generation. *MiraData* features videos with longer durations and structured captions, providing a rich and diverse resource for training models capable of generating extended video sequences with enhanced motion and coherence.

# 3 MiraData Dataset

*MiraData* is a large-scale text-video dataset with long duration and structured detailed captions. We show the overview of the collection and annotation pipeline of *MiraData* in Fig. 1. The final dataset was obtained through a five-step process, which involved collection (in Sec. 3.1), splitting and stitching (in Sec. 3.2), selection (in Sec. 3.3), and captioning (in Sec. 3.4).

## 3.1 Data Collection

The source of videos is crucial in determining the dataset's data distribution. In video generation tasks, there are typically four key expectations: (1) diverse content, (2) high visual quality, (3) long duration, and (4) large motion strength. Existing text-to-video datasets [11, 12, 42] mainly consist of videos from YouTube. Although YouTube offers a vast collection of diverse videos, a large proportion of the videos lack the necessary aesthetic quality for video generation needs. To address all four aspects simultaneously, we select source videos from YouTube, Videvo, Pixabay, and Pexels [2], ensuring a more comprehensive and suitable data source for video generation tasks.

**YouTube Videos.** Following previous works [12, 11, 42], we include YouTube as one of the video sources. However, prior research mainly focuses on collecting diverse videos that are suitable for understanding tasks while giving limited consideration to the need for generation tasks (*e.g.,* duration, motion strength, and visual quality), which are crucial for learning physical laws and 3D consistency.

To address these limitations, we manually select 156 high-quality YouTube channels that are suitable for generation tasks. These channels encompass various categories with rich motion and long video clips, including (1) 3D engine-rendered scenes, (2) city/scenic tours, (3) movies, (4) first-person perspective camera videos, (5) object creation/physical law demonstrations, (6) timelapse videos, and (7) videos showcasing human motion. We collect around $68K$ videos with 720p resolution from these YouTube channels ($K$ denotes thousand). After the video

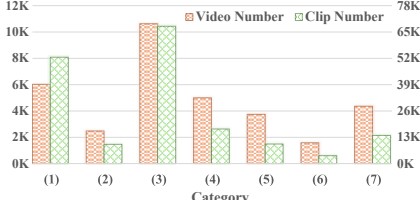

Figure 2: **The video and video clip distribution of different video categories.** (1) to (7) is explained in Sec. 3.1.

splitting and stitching operation described in Sec. 3.2, we obtain around $34K$ videos with $173K$ video clips. The number of videos and clips for each category are shown in Fig. 2. We collect more videos from 3D engine-rendered scenes and movies because they exhibit greater diversity and better

---

[2]YouTube: `https://www.youtube.com/`, Videvo: `https://pixabay.com/`, Pixabay: `https://www.videvo.net/`, Pexels: `https://www.pexels.com/`

visual quality. Moreover, the simplicity and consistency of the physical laws in 3D engine-rendered videos are crucial for enabling video generation models to learn and understand physical laws.

Additionally, to ensure data diversity and amount, we also include videos from HD-VILA-100M [12]. Although this dataset contains around 100 million video clips, after the splitting and stitching operation in Sec. 3.2, only $195K$ clips remain. This further demonstrates the quality of our selected video sources, as evidenced by a higher retention rate considering video duration and continuity.

**Videvo, Pixabay, and Pexels Videos.** These three websites offer stock videos and motion graphics free from copyright issues, which are usually exceptionally high-quality videos uploaded by skilled photographers. Although the videos are usually shorter in duration compared to YouTube, they can compensate for the deficiencies in the visual quality of YouTube videos. Therefore, we collect and annotate videos from these websites, which can enhance the generated videos' aesthetics. We finally obtain around $63K$ videos from Videvo, $43K$ videos from Pixabay, and $318K$ videos from Pexels.

## 3.2 Video Splitting and Stitching

An ideal video clip for video generation should have semantically coherent content, either without shot transitions or with strong continuity between transitions. To achieve this, we conduct a two-stage splitting and stitching process on YouTube videos. In the splitting stage, we use shot change detection with a low threshold to divide the video into segments[3] , ensuring that all distinct clips are extracted. We then stitch short clips together to avoid incorrect separation, considering content-coherent video transitions and accuracy. We employ Qwen-VL-Chat[55], LLaVA[56, 57], ImageBind[58], and DINOv2[59] to assess whether adjacent short clips should be connected. Vision language models excel in detecting content-coherent transitions, while image feature cosine similarity is more effective in connecting incorrect separations. A connection is made only if both vision language models or both image feature extraction models agree. We retain clips longer than 40 seconds for *MiraData*. Since Videvo, Pixabay, and Pexels videos are naturally in clip form, we select clips longer than 10 seconds to filter for longer videos with greater motion strength. Fig. 3 presents the distribution of video clip duration from YouTube and other sources.

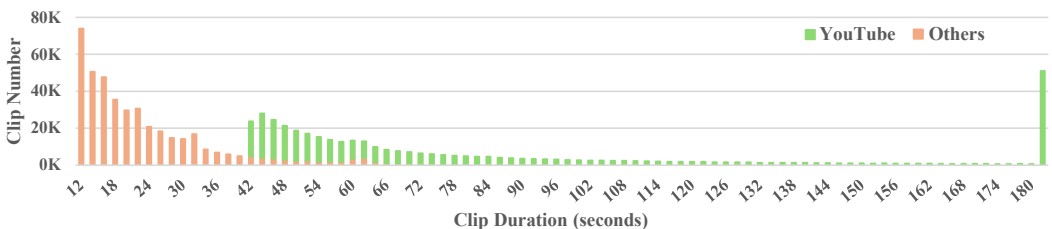

Figure 3: **Distribution of video clip duration from YouTube and other sources**.

## 3.3 Video Selection

*MiraData* provides 5 data versions with different quality levels for video generation training, filtered using four criteria: (1) Video Color, (2) Aesthetic Quality, (3) Motion Strength, and (4) Presence of NSFW Content. For Video Color, we filter videos shot in overly bright or dark environments by calculating average color and the color of the brightest and darkest 80% of frames. Aesthetic Quality is assessed using the Laion-Aesthetic[40] Aesthetic Score Predictor. Motion Strength is measured using the RAFT[60] algorithm to calculate optical flow between frames. NSFW content is detected using the Stable Diffusion Safety Checker [18] on 8 evenly selected frames per video. For criteria (1)-(3), we standardize the frame rate to 2 fps and filter videos into four lists based on increasing threshold values. NSFW videos are filtered out from all datasets. The 5 filtered versions contain 788K, 330K, 93K, 42K, and 9K video clips. Details about the filtering process and thresholds are in the supplementary files.

---

[3]We use PySceneDetect content-aware detection with a threshold of 26

## 3.4 Video Captioning

As emphasized by PixArt[4] and DALL-E 3[20], the quality and granularity of captions are crucial for text-to-image generation. Given the similarities between image and video generation, detailed and accurate textual descriptions should also play a vital role in the latter. However, previous video-text datasets with meta-information annotations (e.g., WebVid-10M[10], HD-VILA-100M[12]) often have incorrect temporal alignment or inaccurate descriptions. Current state-of-the-art video captioning methods generate either simple (e.g., Panda-70M[11]) or inaccurate (e.g., Video-LLaVA[61]) captions. To obtain detailed and accurate captions, we use the more powerful GPT-4V [62], which outperforms existing open-source methods.

To enable GPT-4V, a vision language model with image input only, to understand videos, we extract 8 uniformly sampled frames from each video and arrange them in a $2 \times 4$ grid within a single image. This approach reduces computational cost and facilitates accurate caption generation. Following DALL-E 3[20], we bias GPT-4V to produce video descriptions useful for learning a text-to-video generation model. We first use Panda-70M[11] to generate a "short caption" describing the main subject and actions, which serves as an additional hint for GPT-4V. The GPT-4V-generated "dense caption" covers the main subject, movements, style, backgrounds, and cameras.

To obtain more detailed, fine-grained, and accurate captions, we propose the use of structured captions. In addition to the short and dense captions, structured captions provide further descriptions of crucial elements in the video, including: (1) Main Object: describes the primary object or subject in the video, capturing their attributes, actions, positions, and movements, (2) Background: provides context about the environment or setting, including objects, location, weather, and time, (3) Camera Movements: details any camera pans, zooms, or other movements, and (4) Video Style: covers the artistic style, as well as the visual and photographic features of the video (*e.g.*, realistic, cyberpunk, and cinematic). Thus, each video in MiraData is accompanied by six types of captions: short caption, dense caption, main object caption, background caption, camera caption, and style caption. This creates a hierarchical structure, progressing from a general overview to a more detailed description.

These structured captions provide extra detailed descriptions from various perspectives, enhancing the richness of the captions. With our carefully designed prompt, we can efficiently obtain the video's structured caption from GPT-4V in just one conversation round. As demonstrated in Tab. 1 and Fig. 4, the average caption length of dense descriptions and structured captions has significantly increased to 90 and 214 words respectively, greatly enhancing the descriptive capacity of the captions.

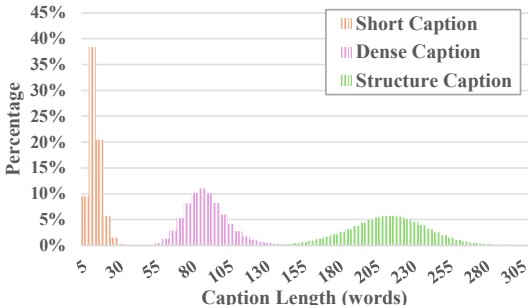

Figure 4: **Distribution of caption length.**

## 3.5 Comparison on Numerical Statistics

We calculate the average frame optical flow strength and aesthetic score on *MiraData*'s unfiltered version ($788K$ video clips) and filtered version ($330K$ video clips) with previous video generation datasets (Panda-70M [11], HD-VILA-100M [12], InternVid [41], and WebVid-10M [10]). For *MiraData*, we calculated the metrics on the full dataset. For other datasets, we randomly select $10K$ video clips to save computation costs. The frame rate is standardized to 2 for both metrics. The results in Tab. 2 show the superiority of *MiraData*, considering both visual quality and motion strength.

Table 2: **Numerical statics comparison of previous datasets and *MiraData*.**

| Metrics | Panda-70M | HD-VILA-100M | InternVid | WebVid-10M | **MiraData**$_{unfilter}$ | **MiraData**$_{filter}$ |
|---|---|---|---|---|---|---|
| **Optical Flow** ↑ | 4.37 | 4.45 | 3.92 | 1.08 | 5.22 | **6.93** |
| **Aesthetic Score** ↑ | 4.67 | 4.61 | 4.50 | 4.41 | 5.01 | **5.02** |

## 4 MiraBench

### 4.1 Prompt Selection

Following EvalCrafter [63], we propose four categories: human, animal, object, and landscape. We randomly select 400 video captions, manually curate them for balanced representation across meta-classes, and prioritize captions closely matching the original videos. We select 50 precise video-text pairs, using short, dense, and structured captions as prompts, forming a set of 150 prompts.

### 4.2 Metrics Design

We design 17 evaluation metrics in *MiraBench* from 6 perspectives, including temporal consistency, temporal motion strength, 3D consistency, visual quality, text-video alignment, and distribution consistency. These metrics encompass most of the common evaluation standards used in previous video generation models and text-to-video benchmarks. Compared to previous benchmarks like VBench [64], our metrics place more emphasis on the model's performance with general prompts instead of manually designed prompts and emphasize 3D consistency and motion strength.

**Temporal Motion Strength.** (1) *Dynamic Degree.* Following previous works [64, 41], we use the average distance of optical flow estimated by RAFT [60] to estimate the dynamics degree. (2) *Tracking Strength.* In optical flow, the objective is to estimate the velocity of all points within a video frame. This estimation is performed jointly for all points, but the motion is predicted only at an infinitesimal distance. In tracking, the goal is to estimate the motion of points over an extended period. Therefore, the distance of tracking points can better distinguish whether the video involves long-range or minor movements (*e.g.*, camera shake or local movements that move back and forth). As shown in Fig. 5 (a), the left figure exhibits a smaller motion distance than the right. However, in Fig. 5 (b), the dynamic degree is incorrectly 1.2 for the left and 0.7 for the right, suggesting that the left motion is larger. Tracking strength in Fig. 5 (c) accurately reflects the moving distance, with 4.1 for the left and 11.8 for the right. We use CoTracker [65] to calculate the tracking path and average the tracking points' distance from the initial frame as the tracking strength metric.

Figure 5: **Illustration of the difference between tracking strength and optical flow dynamic degree.** *Best viewed with Acrobat Reader. Click the images to play the animation clips.*

**Temporal Consistency.** (3) *DINO (Structural) Temporal Consistency.* DINO [59] focuses on structural information. We calculate the cosine similarity of adjacent frames' DINO features to assess structural temporal consistency. (4) *CLIP (Semantic) Temporal Consistency.* We calculate the cosine similarity of adjacent frames' CLIP [13] features to assess structural temporal consistency since CLIP focuses on semantic information. (5) *Temporal Motion Smoothness.* Following VBench [64], we use the motion priors in the video interpolation model AMT [66] to calculate the motion smoothness. Since larger motion is expected to contain smaller consistency and vice versa, we multiply *Tracking Strength* by these feature similarities to obtain more reasonable temporal consistency metrics.

**3D Consistency.** Following GVGC [67], we calculate (6) *Mean Absolute Error*, and (7) *Root Mean Square Error* to evaluate video 3D consistency from the perspective of 3D reconstruction.

**Visual Quality.** (8) *Aesthetic Quality*. We evaluate the aesthetic score of generated video frames using the LAION aesthetic predictor [18]. (9) *Imaging Quality*. Following VBench [64], we evaluate video distortion (*e.g.*, over-exposure, noise, and blur) using the MUSIQ [68] quality predictor.

268 **Text-Video Alignment.** We use ViCLIP [41] to evaluate the consistency between video and text. We
269 calculate from 5 aspects following *MiraBench* prompt structure: (10) ***Camera Alignment***. (11) ***Main***
270 ***Object Alignment***. (12) ***Background Alignment***. (13) ***Style Alignment***. (14) ***Overall Alignment***.

271 **Distribution Similarity.** Following previous works [3, 23, 54], we use (15) ***FVD*** [69], (16) ***FID*** [70],
272 (17) ***KID*** [71] to evaluate the distribution similarity of generated and training data.

## 5 Experiments

### 5.1 Model Design of MiraDiT

275 To validate the effectiveness of MiraData for consistent long-video generation, we design an efficient
276 pipeline based on Diffusion Transformer [72], as illustrated in Fig.6. Following SVD [2], we use a
277 hybrid Variational Autoencoder with a 2D convolutional encoder and a 3D convolutional decoder to
278 reduce flickering in generated videos. Unlike previous methods[2, 34, 33] that rely on short captions
279 and typically use a CLIP text encoder with 77 output tokens, we employ a larger Flan-T5-XXL [73]
280 for textual encoding, supporting up to 512 tokens for dense and structured caption understanding.

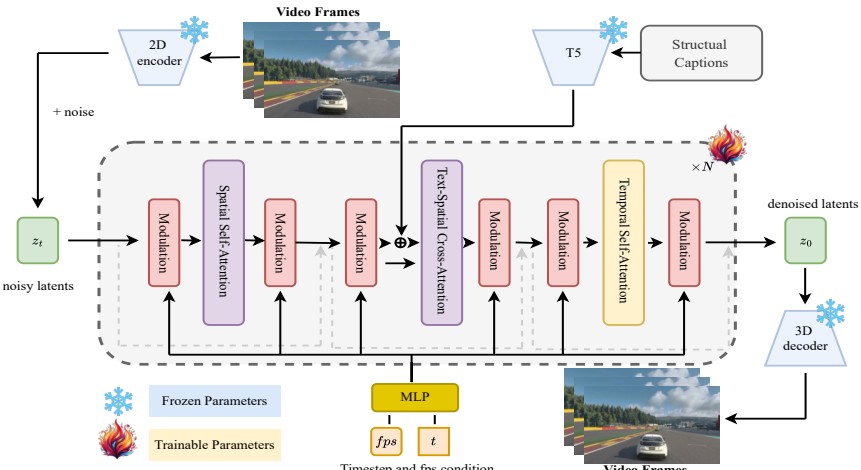

Figure 6: **MiraDiT pipeline for long video generation.**

281 **Text-spatial cross-attention.**  For latent denoising, we build a spatial-temporal transformer as the
282 trainable generation backbone. As shown in Fig.6, we adopt spatial and temporal self-attention
283 separately rather than full attention on all video pixels to reduce the heavy computational load of
284 long-video generation. Similar to W.A.L.T [26], we apply extra conditioning on spatial queries during
285 cross-attention to stabilize training and improve generation performance. For faster convergence, we
286 partially initialize spatial attention layers from weights of text-to-image model Pixart-alpha [4], while
287 keeping other layers trained from scratch.

288 **FPS-conditioned modulation.**  Following DiT and Stable Diffusion 3 [6], we use a modulation
289 mechanism for the current timestep condition. Additionally, we embed an extra current FPS condition
290 in the AdaLN layer to enable motion strength control during inference in the generated videos.

291 **Dynamic frame length and resolution.**  We train MiraDiT in a way that supports generating videos
292 with different resolutions and lengths to evaluate the model performance on motion strength and
293 3D consistency in different scenarios. Inspired by NaViT [74], which uses Patch n' Pack to achieve
294 dynamic resolution training, we apply a Frame n' Pack strategy to train videos with various temporal
295 lengths. Specifically, we randomly drop frames with zero padding using a temporal mask, then apply
296 masked self-attention and positional embeddings according to the temporal masks. The gradients of
297 masked frames are stopped as well. However, for varying resolution training, we didn't adopt Patch

298 n' Pack since it made the model harder to train during our early experiments. Instead, we follow
299 Pixart [4] and use a bucket strategy where the models are trained on different resolution videos where
300 each training batch only contains videos of the same resolution.

301 **Inference details.** During inference, we use the DDIM [75] sampler with 25 steps and classifier-
302 free guidance of scale 12. The fps condition can be set between 5 and 30, allowing for flexibility in
303 the generated video's frame rate. For evaluation purposes, we test all our models at 6 fps to ensure a
304 consistent comparison across different settings. To further enhance the visual quality of the generated
305 videos, we provide an optional post-processing step using the RIFE [76] model. By applying $4\times$
306 frame interpolation, we can increase the frame rate of the generated video to 24 fps, resulting in
307 smoother motion and improved overall appearance.

## 5.2 Comparison with Previous Video Generation Datasets

309 Our experiments aim to validate the effectiveness of MiraData in long video generation by assessing
310 (1) temporal motion strength and consistency, and (2) visual quality and text alignment. We train
311 MiraDiT models on WebVid-10M and MiraData separately, evaluating them on MiraBench at
312 $384 \times 240$ resolution with 5s length using 14 metrics covering motion strength, consistency, visual
313 quality, and text-video alignments.

314 Tab. 2 shows that the model trained on MiraData demonstrates significant improvements in motion
315 strength while maintaining temporal and 3D consistency compared to the WebVid-10M model.
316 Moreover, MiraData's higher-quality videos and dense, accurate prompts lead to better visual quality
317 and text-video alignments in the trained model. We compare our MiraDiT model trained on MiraData
318 to state-of-the-art open-source methods, OpenSora [77] (DiT-based) and VideoCrafter2 [35] (U-Net-
319 based). Our model significantly outperforms previous methods in terms of motion strength and 3D
320 consistency while achieving competitive results in visual quality and text-video alignment. This
321 demonstrates MiraData's effectiveness in enhancing long video generation. Note that distribution-
322 based metrics like FVD are not reported due to the difference in training datasets. More visual and
323 metric comparisons are in the Appendix.

Table 3: **Comparison of MiraDiT trained on MiraData and WebVid-10M [10].** $\uparrow$ and $\downarrow$ means
higher/lower is better. 1) - 14) indicates indices of metrics in MiraBench (Sec. 4), where DD for
Dynamic Degree, TS for Tracking Strength, DTC for DINO Temporal Consistency, CTC for CLIP
Temporal Consistency, TMS for Temporal Motion Smoothness, MAE for Mean Absolute Error,
RMSE for Root Mean Square Error, AQ for Aesthetic Quality, IQ for Imaging Quality, CA for
Camera Alignment, MOA for Main Object Alignment, BA for Background Alignment, SA for Style
Alignmnet, and OA for Overall Alignment. Best shown in **blod**, and second best shown in underlined.

| Metrics | Temporal Motion Strength | | Temporal Consistency | | | 3D Consistency | |
|---|---|---|---|---|---|---|---|
| | 1) DD$_\uparrow$ | 2) TS$_\uparrow$ | 3) DTC$_\uparrow$ | 4) CTC$_\uparrow$ | 5) TMS$_\uparrow$ | 6) MAE$_{\downarrow \times 10^{-2}}$ | 7) RMSE$_{\downarrow \times 10^{-1}}$ |
| OpenSora [77] | 7.65 | 16.07 | 12.34 | 13.20 | 13.70 | **75.45** | **10.39** |
| VideoCrafter2 [35] | 1.71 | 6.72 | 6.41 | 6.36 | 6.60 | 101.55 | 13.05 |
| MiraDiT (WebVid-10M [10]) | 7.12 | 22.36 | 20.24 | 20.97 | 21.86 | 91.48 | 12.11 |
| MiraDiT (*MiraData*) | **15.46** | **49.47** | **43.78** | **45.95** | **47.24** | 85.27 | 11.74 |

| Metrics | Visual Quality | | Text-Video Alignmnet | | | | |
|---|---|---|---|---|---|---|---|
| | 8) AQ$_{\uparrow \times 10^-}$ | 9) IQ$_\uparrow$ | 10) CA$_\uparrow$ | 11) MOA$_\uparrow$ | 12) BA$_\uparrow$ | 13) SA$_\uparrow$ | 14) OA$_\uparrow$ |
| OpenSora [77] | 47.10 | 59.54 | 12.40 | **18.12** | **13.20** | **13.35** | 16.12 |
| VideoCrafter2 [35] | **58.69** | **64.96** | 12.00 | 17.90 | 11.25 | 12.15 | **16.90** |
| MiraDiT (WebVid-10M [10]) | 43.11 | 58.58 | 12.35 | 14.32 | 11.90 | 12.32 | 15.31 |
| MiraDiT (*MiraData*) | 49.90 | 63.71 | **12.66** | 14.67 | 12.18 | 12.59 | 16.66 |

324 To provide a more comprehensive assessment, we present the human evaluation results in Tab. 4.
325 We enlisted 6 volunteers to evaluate the entire validation set of MiraBench. Each volunteer was
326 provided with a set of 4 videos generated using OpenSora [77], VideoCrafter2 [35], MiraDiT trained
327 on WebVid-10M [10], and MiraDiT trained on MiraData. The evaluators were asked to rank the four
328 videos from best to worst (1-4) based on five criteria: (1) motion strength, (2) temporal consistency,
329 (3) 3D consistency, (4) visual quality, and (5) text-video alignment. We observe that there are some

alignments and discrepancies between human evaluation (Tab. 4) results and automatic evaluation results (Tab. 3), and explain for the discrepancies here: (1) For the Temporal Consistency metric in the automatic evaluation, we multiply Tracking Strength by the feature similarities among adjacent video frames. This approach ensures that the metric does not unfairly favor static videos, which would naturally achieve the highest temporal consistency due to their lack of motion. However, in human evaluations, it is challenging to have annotators consider both metrics simultaneously. Therefore, we simply ask the question "Is this video temporally consistent?". This make methods like VideoCrafter receiving high human evaluation scores, as the videos generated by VideoCrafter exhibit very low motion strength. (2) For 3D consistency metric, we find it hard for human beings to accurately judge whether a video's scene is 3D consistency (e.g., alignment with 3D modeling standards and physical optics projection). However, automatic metrics also face difficulties due to unignorable calculation errors in 3D modeling methods. Therefore, we believe that the most effective approach is to incorporate both automated and human indicators in the evaluation process.

Table 4: **Human evaluation results** of MiraDiT trained on MiraData and WebVid-10M [10], as well as open-source methods, OpenSora (DiT-based) [77] and VideoCrafter2 (U-Net-based) [35].

| Metrics | Motion Strength ↓ | Temporal Consistency ↓ | 3D Consistency ↓ | Quality ↓ | Text Alignment ↓ |
|---|---|---|---|---|---|
| OpenSora [77] | 2.6 | 2.5 | 2.6 | 2.8 | 2.9 |
| VideoCrafter2 [35] | 2.9 | **1.8** | 2.3 | **1.4** | 2.3 |
| MiraDiT (WebVid-10M [10]) | 3.2 | 3.8 | 3.0 | 3.5 | 2.7 |
| MiraDiT (MiraData) | **1.3** | 1.9 | **2.1** | 2.3 | **2.1** |

## 5.3   Role of Caption Length and Granularity

We investigate the impact of caption length and granularity on MiraDiT's performance by evaluating the model using short, dense, and structural captions separately. The results in Tab. 5 demonstrate that longer and more detailed captions do not necessarily improve the visual quality of the generated videos. However, they offer significant benefits in terms of increased dynamics, enhanced temporal consistency, more accurate generation control, and better alignment between the text and the generated video content. These findings highlight the importance of caption granularity in guiding the model to produce videos that more closely match the desired descriptions while maintaining coherence and realism. Please see appendix for more qualitative results and detailed ablation studies.

Table 5: **Comparison of MiraDiT model with different caption length and granularity.** 1) - 14) indicates indices of metrics in MiraBench (Sec. 4). See Tab. 3 for the meaning of metrics annotation.

| Metrics | 1) DD↑ | 2) TS↑ | 3) DTC↑ | 4) CTC↑ | 5) TMS↑ | 8) AQ↑ | 9) IQ↑ | 14) OA↑ |
|---|---|---|---|---|---|---|---|---|
| Short Caption | 9.45 | 27.03 | 24.39 | 25.20 | 26.05 | 4.84 | 63.64 | 7.73 |
| Dense Caption | 17.39 | 52.53 | 46.13 | 48.35 | 50.12 | 5.14 | 63.43 | 14.88 |
| Structural Caption | 19.53 | 68.85 | 60.83 | 64.31 | 65.56 | 4.99 | 64.07 | 15.36 |

## 6   Conclusion and Discussion

**Conclusion.** In conclusion, *MiraData* complements existing video datasets with high-quality, long-duration videos featuring detailed captions and strong motion intensity. Curated from diverse video sources and annotated with multiple high-performance models, *MiraData* shows advantages in comprehensive evaluation framework *MiraBench* with the designed *MiraDiT* model, highlighting its potential to push the boundaries of high-motion, temporally consistent long video generation.

**Limitation.** Despite *MiraData*'s advantages over previous datasets, it still has limitations, such as inherent biases, potential annotation errors, and insufficient coverage. The evaluation metrics in *MiraBench* may also yield inaccurate results in uncommon video scenarios, such as jitter or overexposure. Due to the page limit, the appendix will provide a detailed discussion.

**Potential Negative Societal Impacts.** The enhanced video generation capabilities promoted by *MiraData* could lead to negative societal impacts and ethical issues, including the creation of deepfakes and misinformation, privacy breaches, and harmful content generation. We would engage in implementing stringent ethical guidelines, ensuring robust privacy protections, and promoting unbiased dataset curation to prevent these issues. The appendix provides a detailed discussion.

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
