—— **Appendix** ——
# MiraData: A Large-Scale Video Dataset with Long Durations and Structured Captions

**Xuan Ju**[1,2*], **Yiming Gao**[1*], **Zhaoyang Zhang**[1†*], **Ziyang Yuan**[1], **Xintao Wang**[1],
**Ailing Zeng**[2], **Yu Xiong**[2], **Qiang Xu**[2], **Ying Shan**[1]
`https://github.com/mira-space/MiraData`

In the appendix, we first give out more details about the collection, selection, and annotation of *MiraData* in Sec. A. Then, we provide additional experiment results of quantitative comparison, qualitative comparison, and ablation in Sec. B. In Sec. C, we further explain the limitations, societal impact, ethical issues and broad impact of our dataset. Finally, in Sec. D, we provide data acquisition, data documentation, and data license for ease of data use.

## A   MiraData: Additional Details

### A.1   Data Collection

We provide additional details about collecting YouTube video channels in this section. We select 7 categories that contain more rich motion and long video clips: (1) 3D engine-rendered scenes, (2) city/scenic tours, (3) movies, (4) first-person perspective camera videos, (5) object creation/physical law demonstrations, (6) timelapse videos, and (7) videos showcasing human motion.

The reason we choose these channels is as follows:

(1) For **3D engine-rendered scenes**, the videos are typically recorded in 3D rendering engines with predefined physics laws. Thus, they often contain rich scene and perspective changes, with relatively long continuous shots, making them suitable for learning long video generation.

(2) **City/scenic tours** are usually filmed by people walking with handheld cameras in urban or scenic areas. Consequently, the scenes are relatively continuous and possess strong 3D spatial descriptive capabilities.

(3) **Movies** usually contain high-quality visuals and seamless transitions in the same scene, allowing for a more comprehensive description of the same scene from different angles.

(4) **First-person-perspective camera videos** provide a perspective from the vantage point of the person or device capturing the footage. Compared to city/scenic tours, this category focuses more on extreme sports and typically uses camera lenses with slight distortion, which offers a view from the eyes of the subject.

(5) **Object creation/physical law demonstration** often includes demonstrative videos focused on a single perspective, such as baking tutorials or explanations of physical principles. Due to their relatively simple scenes and clear procedural steps, these videos are beneficial for learning physical laws in long videos.

(6) **The timelapse videos** capture a sequence of images at set intervals to record changes that take place slowly over time, which represent processes that would be too slow to observe in

---
*Equal contribution. † Project Lead. [1]ARC Lab, Tencent PCG. [2]The Chinese University of Hong Kong.

Submitted to the 38th Conference on Neural Information Processing Systems (NeurIPS 2024) Track on Datasets and Benchmarks. Do not distribute.

32  real-time. This would be helpful for the video generation model to learn real-world physics
33  knowledge as indicated by MagicTime [1].

(7) **Human motion videos** show human movements, such as speeches, dances, and model stage
35  performances. Including this category will be beneficial for generating long videos that
36  include localized limb movements of the human body. In Fig. 1, we provided two examples
37  from each category to illustrate the differences between the various types of videos.

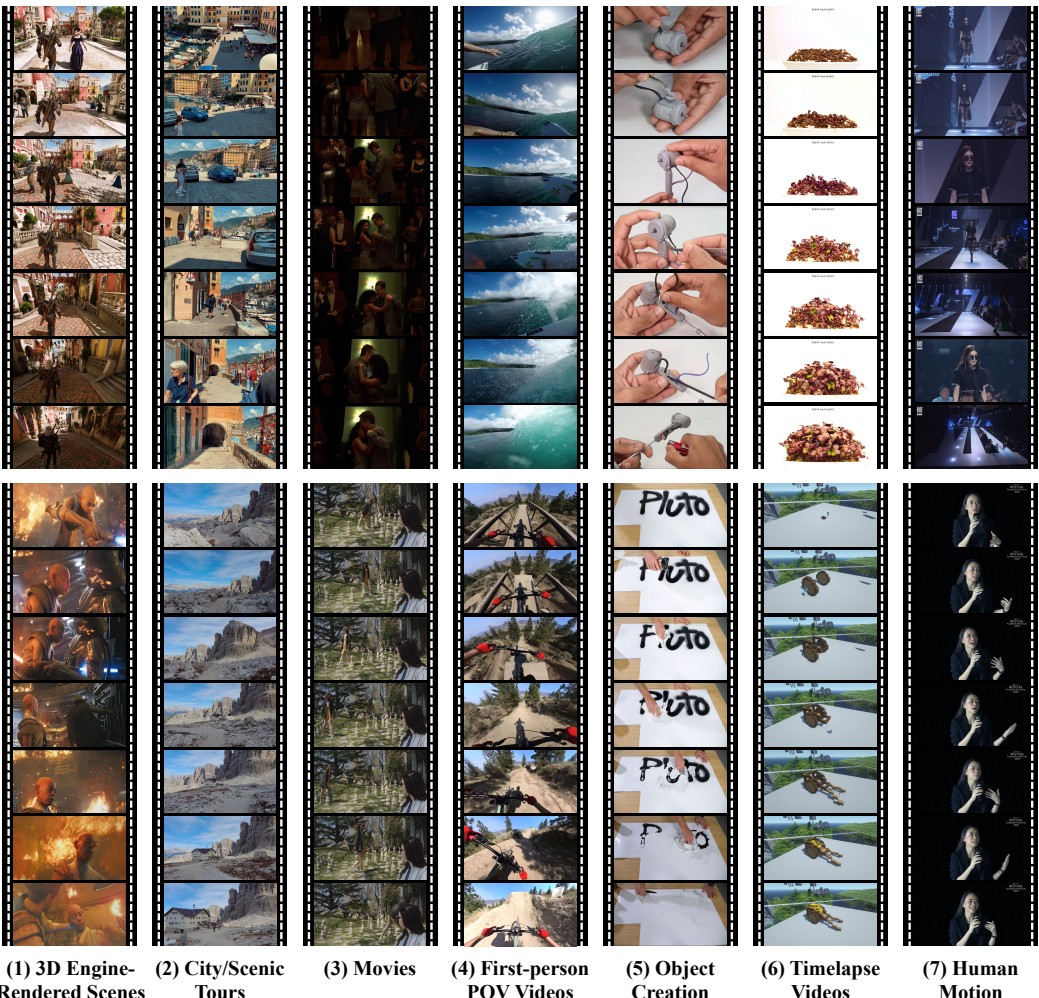

**(1) 3D Engine-Rendered Scenes**    **(2) City/Scenic Tours**    **(3) Movies**    **(4) First-person POV Videos**    **(5) Object Creation**    **(6) Timelapse Videos**    **(7) Human Motion**

Figure 1: **Video Examples From Each Category.**

## A.2   Video Splitting and Stitching

39  For video splitting, we use PySceneDetect[2] content-aware detection with a threshold of 26. This
40  process may result in some incorrect separations when cutting long videos into small clips. To address
41  this issue, we consider both content-coherent video transitions and wrong cuts.

42  To connect content-coherent video clips, we employ Qwen-VL-Chat[2] and LLaVA[3]. For each
43  pair of adjacent video clips, we extract the 5th frame from the end of the former video and the 5th
44  frame from the beginning of the latter video. These two frames are concatenated and input into the
45  language models with the following prompt:

---

[2]https://www.scenedetect.com/

"Given two images shown on the right and the left, please determine whether the two images are similar to each other and coming from the same video. Please answer 'Yes' or 'No'. You can start by examining the visual content of the two images. Look for similarities in various aspects, such as main objects, backgrounds, colors, lighting conditions, and spatial arrangements. Consider both global and local features within the images. For example, you should output 'Yes' for two images from different views of a scene. You should output 'No' for two unrelated images."

The language models will output "Yes" or "No" as the answer. The two adjacent clips will be connected only when both models output "Yes". To connect wrong cuts, we use ImageBind[4] and DINOv2[5] with thresholds of $0.6$ and $0.85$, respectively.

## A.3   Video Selection

We list the filtering criteria in Tab. 1. Average Optical flow measures the overall motion across the video sequence, giving an idea of how much movement is occurring on average. Image Max 30% Optical Flow identifies each image's maximum 30% optical flow values. This can focus on the part of the image that contains the largest movement. Temporal Min 30% Optical Flow the minimum optical flow values within the bottom 30% of frames in terms of motion intensity, giving the results of the least dynamic parts of the image sequence by focusing on the frames with the lowest motion. The Average Aesthetic Score is assessed using the Laion-Aesthetic[6] Aesthetic Score Predictor and averaged among frames. Average Color is the average of the color of every pixel in frames. Temporal Max 80% Color identifies the maximum color values within the top 80% of frames, which is the brightest. Temporal Min 80% Color identifies the maximum color values within the bottom 80% of frames, which is the darkest. Contain NSFW identifies whether the frames contain NSFW content.

Table 1: **Filtering Criteria of *MiraData*.** We offer five versions of MiraData, each filtered using different criteria to cater to various research needs and preferences.

| Metrics | 788K Version | 330K Version | 93K Version | 42K Version | 9K Version |
|---|---|---|---|---|---|
| Average Optical Flow | - | >2.0 | >3.0 | >4.0 | >4.5 |
| Image Max 30% Optical Flow | - | - | >4.3 | >4.8 | >5.1 |
| Temporal Min 30% Optical Flow | - | - | >2.5 | >3.5 | >4.0 |
| Average Aesthetic Score | - | - | >3.0 | >5.0 | >5.5 |
| Average Color | - | >25.0 <230.0 | >25.0 <230.0 | >35.0 <220.0 | >35.0 <220.0 |
| Temporal Max 80% Color | - | - | <235.0 | <225.0 | <225.0 |
| Temporal Min 80% Color | - | - | >20.0 | >30.0 | >30.0 |
| Contain NSFW | No | No | No | No | No |

## A.4   Video Caption

To facilitate the comprehension of videos by GPT-4V, we extract eight uniformly sampled frames from each video, arranging them in a $2 \times 4$ grid within a single image. Alongside this $2 \times 4$ grid image, we meticulously design a prompt to enable GPT-4V to perceive this image as a video thumbnail. Following DALL-E3 [7], we bias GPT-4V to yield video descriptions conducive to the learning of a text-to-video generation model. Our initial step utilizes Panda-70M [8] to produce a "short caption" that delineates the primary subject and actions, serving as an additional cue for GPT-4V. Specifically, our prompt begins with the following guiding content:

> A wide image is given containing a $2 \times 4$ grid of 8 equally spaced video frames. They're arranged chronologically from left to right, and then from top to down, all separated by white borders. This video depicts "*Short Captions*". Please imagine the video based on the sequence of 8 frames, and provide a faithfully concise description of the following content:

We further instruct GPT-4V to generate dense descriptions of videos. In addition, we introduce structured captions to obtain more intricate information. To procure more precise, detailed, and fine-grained structured captions, we carefully craft prompts that inquire about various aspects of the video, including the main object, background, camera movement, and video style. The specific prompts are described below:

1. Detailed description of this video in more than three sentences. Here are some examples of good descriptions: 1) A stylish woman walks down a Tokyo street filled with warm glowing neon and animated city signage. She wears a black leather jacket, a long red dress, and black boots, and carries a black purse. She wears sunglasses and red lipstick. She walks confidently and casually. The street is damp and reflective, creating a mirror effect of the colorful lights. Many pedestrians walk about. 2) A movie trailer featuring the adventures of the 30 year old space man wearing a red wool knitted motorcycle helmet, blue sky, salt desert, cinematic style, shot on 35mm film, vivid colors.

2. Description of the main subject actions or status sequence. This suggests including the main subjects (person, object, animal, or none) and their attributes, their action, their position, and movements during the video frames.

3. Summary of the background. This should also include the objects, location, weather, and time.

4. Summary of the view shot, camera movement and changes in shooting angles in the sequence of video frames.

5. Briefly one-sentence Summary of the visual, Photographic and artistic style.

No need to provide summary content. Do not describe each frame individually. Do not reply with words like 'first frame'. The description should be useful for AI to re-generate the video.

Our carefully-designed prompts enable us to efficiently obtain both dense descriptions and structured captions in a single round of dialogue. This approach minimizes time overhead and computational cost, making it highly effective for generating comprehensive video annotations. Fig. 2 and Fig. 3 visualize the word cloud of short, dense and structured captions respectively.

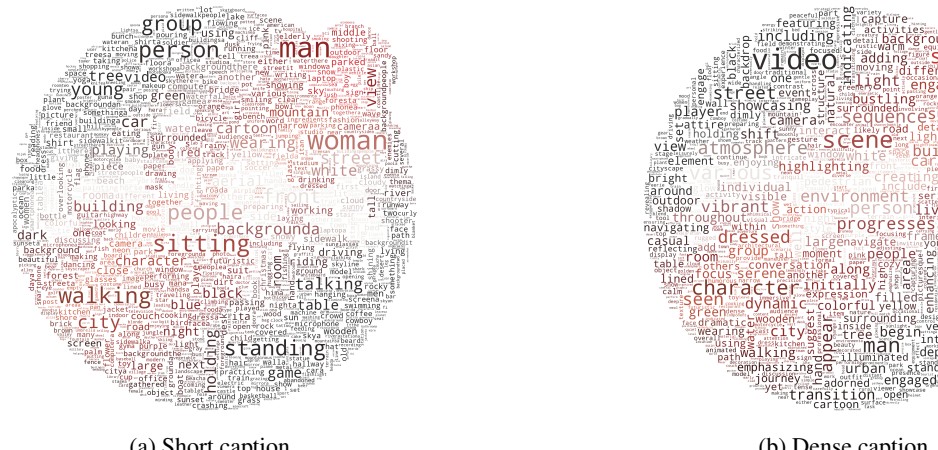

(a) Short caption.                    (b) Dense caption.

Figure 2: The word cloud (Top-2000) of the generated short and dense captions in our *MiraData*, which reveals that our caption highlight main objects and their rich actions.

# B   More Experiment Results

## B.1   Qualitative Comparison

We provide qualitative comparisons of *MiraDiT* trained on *MiraData* and WebVid-10M [9], as well as open-source video generation methods, UNet-based VideoCrafter2 [10] and DiT-based OpenSora [11] in Fig. 4. Results show *MiraDiT* trained on *MiraData* shows much stronger motion than other methods, while other methods show almost static background with limit motion intensity. Comparing with WebVid, our *MiraData* can enable *MiraDiT* to maintain better 3D consistency even with stronger motion intensity.

## B.2   Quantitative Comparison

To report the error bars, we give two more groups of evaluation results with different seeds for the comparison in the main paper, Tab.3, which is shown in Tab. 2. The results with different random seeds show the same trend, where MiraDiT trained on MiraData demonstrates significant

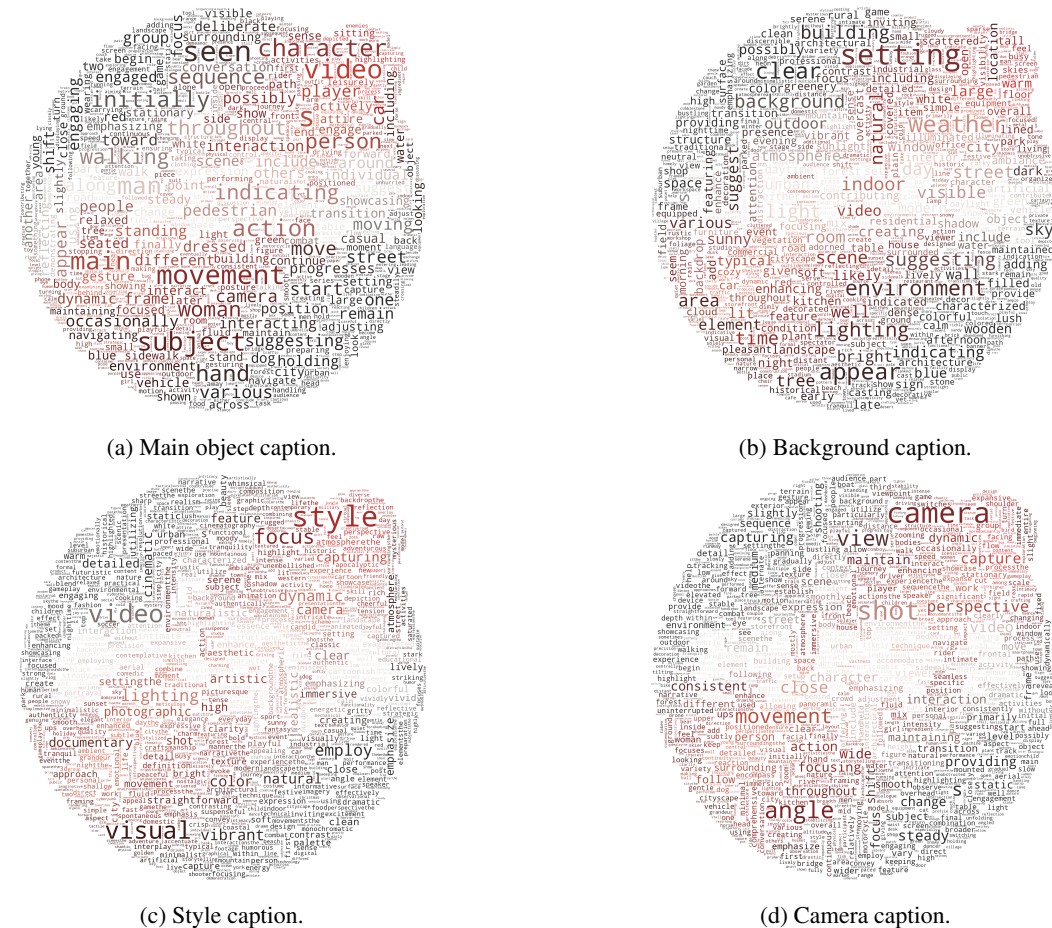

(a) Main object caption.

(b) Background caption.

(c) Style caption.

(d) Camera caption.

Figure 3: The word cloud (Top-2000) of the generated structured captions in our *MiraData*. This reveals that our structured captions can generate accurate corresponding detailed descriptions.

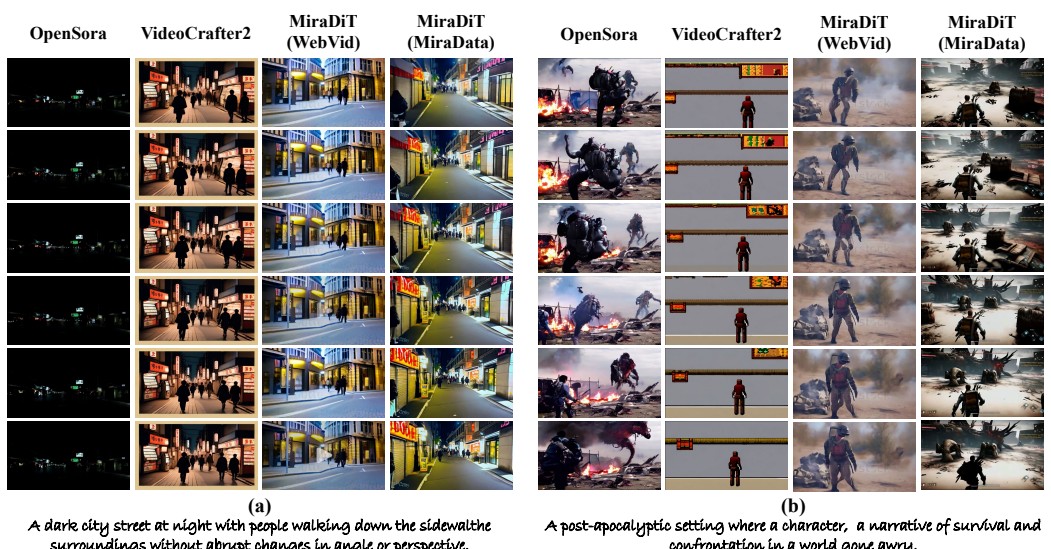

(a)

A dark city street at night with people walking down the sidewalthe surroundings without abrupt changes in angle or perspective.

(b)

A post-apocalyptic setting where a character, a narrative of survival and confrontation in a world gone awry.

Figure 4: **Qualitative comparison of MiraDiT trained on MiraData and WebVid-10M [9], as well as open-source video generation methods. Please refer to the attached video for a better view.**

improvements in motion strength while maintaining temporal and 3D consistency compared to the model trained on WebVid-10M.

Table 2: **Quantitative Comparison of MiraDiT trained on MiraData and WebVid-10M [9].** ↑ and ↓ means higher/lower is better. 1) - 14) are metrics of MiraBench, where DD for Dynamic Degree, TS for Tracking Strength, DTC for DINO Temporal Consistency, CTC for CLIP Temporal Consistency, TMS for Temporal Motion Smoothness, MAE for Mean Absolute Error, RMSE for Root Mean Square Error, AQ for Aesthetic Quality, IQ for Imaging Quality, CA for Camera Alignment, MOA for Main Object Alignment, BA for Background Alignment, SA for Style Alignmnet, and OA for Overall Alignment.

| Metrics | Temporal Motion Strength | | Temporal Consistency | | | 3D Consistency | |
|---|---|---|---|---|---|---|---|
| | 1) DD$_\uparrow$ | 2) TS$_\uparrow$ | 3) DTC$_\uparrow$ | 4) CTC$_\uparrow$ | 5) TMS$_\uparrow$ | 6) MAE$_{\downarrow \times 10^{-2}}$ | 7) RMSE$_{\downarrow \times 10^{-1}}$ |
| OpenSora [11] | 7.65 | 16.07 | 12.34 | 13.20 | 13.70 | 75.45 | 10.39 |
| | 7.48 | 15.21 | 11.86 | 12.94 | 13.03 | 78.23 | 11.06 |
| | 7.59 | 15.78 | 12.01 | 13.04 | 13.42 | 77.18 | 10.82 |
| VideoCrafter2 [10] | 1.71 | 6.72 | 6.41 | 6.36 | 6.60 | 101.55 | 13.05 |
| | 3.01 | 8.52 | 9.12 | 8.89 | 9.23 | 120.05 | 15.13 |
| | 2.02 | 6.91 | 6.53 | 6.42 | 6.84 | 99.84 | 12.87 |
| MiraDiT (WebVid-10M [9]) | 7.12 | 22.36 | 20.24 | 20.97 | 21.86 | 91.48 | 12.11 |
| | 6.93 | 21.74 | 20.23 | 20.49 | 22.30 | 90.11 | 11.96 |
| | 7.18 | 22.52 | 20.30 | 20.99 | 21.95 | 91.31 | 12.12 |
| MiraDiT (*MiraData*) | 15.46 | 49.47 | 43.78 | 45.95 | 47.24 | 85.27 | 11.74 |
| | 15.32 | 49.41 | 43.66 | 45.85 | 47.19 | 84.22 | 11.68 |
| | 16.03 | 50.26 | 44.01 | 45.99 | 47.32 | 86.11 | 11.94 |

| Metrics | Visual Quality | | Text-Video Alignmnet | | | | |
|---|---|---|---|---|---|---|---|
| | 8) AQ$_{\uparrow \times 10^{-}}$ | 9) IQ$_\uparrow$ | 10) CA$_\uparrow$ | 11) MOA$_\uparrow$ | 12) BA$_\uparrow$ | 13) SA$_\uparrow$ | 14) OA$_\uparrow$ |
| OpenSora [11] | 47.10 | 59.54 | 12.40 | 18.12 | 13.20 | 13.35 | 16.12 |
| | 44.28 | 60.14 | 12.38 | 17.93 | 13.41 | 13.39 | 16.82 |
| | 48.01 | 58.39 | 12.01 | 18.25 | 13.38 | 13.96 | 16.57 |
| VideoCrafter2 [10] | 58.69 | 64.96 | 12.00 | 17.90 | 11.25 | 12.15 | 16.90 |
| | 57.96 | 64.86 | 12.09 | 17.86 | 11.63 | 12.09 | 16.59 |
| | 58.28 | 64.99 | 11.97 | 17.78 | 11.42 | 12.04 | 16.68 |
| MiraDiT (WebVid-10M [9]) | 43.11 | 58.58 | 12.35 | 14.32 | 11.90 | 12.32 | 15.31 |
| | 43.01 | 57.74 | 12.43 | 14.29 | 12.01 | 12.29 | 16.01 |
| | 43.42 | 59.00 | 12.42 | 14.33 | 11.96 | 12.21 | 15.48 |
| MiraDiT (*MiraData*) | 49.90 | 63.71 | 12.66 | 14.67 | 12.18 | 12.59 | 16.66 |
| | 50.21 | 63.58 | 12.86 | 14.69 | 12.25 | 12.53 | 16.84 |
| | 49.53 | 63.70 | 12.73 | 14.69 | 12.17 | 12.64 | 16.73 |

To assess MiraDiT's performance on other benchmark datasets, we test the performance of MiraDiT on the recent text-to-video benchmark, T2V-CompBench [12]. T2V-CompBench includes 7 metrics designed to evaluate the alignment of generated videos with the corresponding text prompts: (1) Consistent Attribute Binding: Evaluates whether object attributes remain consistent throughout the generated video frames. (2) Dynamic Attribute Binding: Assesses if the generated video accurately reflects changes in object attributes. (3) Spatial Relationship: Determines if the generated video adheres to the spatial relationships specified in the text prompt. (4) Motion Binding: Assesses the correctness of the object's motion direction in the generated video. (5) Action Binding: Evaluates the accuracy of the object action categories in the generated video. (6) Object Interactions: Tests the model's ability to generate dynamic interactions between objects. (7) Generative Numeracy: Evaluates the accuracy in the number of objects generated as specified in the text prompt. Results show that MiraDiT trained on MiraData achieves much better results on all metrics compare to that trained on WebVid-10M. Moreover, MiraDiT trained on MiraData have the best results on Dynamic Attribute Binding, further illustrates the advantages of training with high-dynamic, detailed-captioned data. MiraDiT trained on MiraData also achieves a relatively advanced results in all open-source text-to-video generation models. However, we must point out, that this comparison is unfair, as different models were trained using different computational resources and distinct models, making it impossible to assess the quality of MiraData relative to other training data. Moreover, the evaluation prompts in T2V-CompBench primarily consist of short captions with only a single simple sentence, which limits MiraData's ability to fully showcase its strengths.

Table 3: **T2V-CompBench evaluation results** of MiraDiT trained on MiraData and WebVid-10M. Best results are shown in bold.

| Method | Consist-attr ↑ | Dynamic-attr ↑ | Spatial ↑ | Motion ↑ | Action ↑ | Interaction ↑ | Numeracy ↑ |
|---|---|---|---|---|---|---|---|
| ModelScope | 0.5483 | 0.1654 | 0.4220 | 0.2552 | 0.4880 | 0.7075 | 0.2066 |
| ZeroScope | 0.4495 | 0.1086 | 0.4073 | 0.2319 | 0.4620 | 0.5550 | 0.2378 |
| Latte | 0.5325 | 0.1598 | 0.4476 | 0.2187 | 0.5200 | 0.6625 | 0.2187 |
| Show-1 | 0.6388 | 0.1828 | 0.4649 | 0.2316 | 0.4940 | **0.7700** | 0.1644 |
| VideoCrafter2 | 0.6750 | 0.1850 | 0.4891 | 0.2233 | 0.5800 | 0.7600 | 0.2041 |
| Open-Sora 1.1 | 0.6370 | 0.1762 | **0.5671** | 0.2317 | 0.5480 | 0.7625 | 0.2363 |
| Open-Sora 1.2 | 0.6600 | 0.1714 | 0.5406 | 0.2388 | 0.5717 | 0.7400 | 0.2556 |
| Open-Sora-Plan v1.0.0 | 0.5088 | 0.1562 | 0.4481 | 0.2147 | 0.5120 | 0.6275 | 0.1650 |
| Open-Sora-Plan v1.1.0 | **0.7413** | 0.1770 | 0.5587 | 0.2187 | **0.6780** | 0.7275 | **0.2928** |
| AnimateDiff | 0.4883 | 0.1764 | 0.3883 | 0.2236 | 0.4140 | 0.6550 | 0.0884 |
| VideoTetris | 0.7125 | 0.2066 | 0.5148 | 0.2204 | 0.5280 | 0.7600 | 0.2609 |
| LVD | 0.5595 | 0.1499 | 0.5469 | **0.2699** | 0.4960 | 0.6100 | 0.0991 |
| MagicTime | - | 0.1834 | - | - | - | - | - |
| MiraDiT (WebVid-10M) | 0.6012 | 0.1972 | 0.4438 | 0.2250 | 0.5156 | 0.6075 | 0.1909 |
| MiraDiT (MiraData) | 0.6825 | **0.2302** | 0.4622 | 0.2321 | 0.6340 | 0.7373 | 0.2234 |

## B.3 Role of Video Duration

To evaluate the effectiveness of *MiraData* on long-duration video generation, we train a dynamic frame rate video generation model that supports arbitrary length video generation from 0 to 20s on *MiraData* and WebVid respectively. Tab. 4 presents the results for 5s, 10s, and 20s videos. Tab. 4 presents the results for 5s, 10s, and 20s videos. The experimental results demonstrate that our *MiraData* achieves significantly better motion strength and dynamic degree compared to the model trained on WebVid-10M, while maintaining consistent temporal and 3D consistency. Furthermore, *MiraData* yields higher aesthetic scores, attributed to its high video visual quality (e.g., resolution and aesthetic scores). As the generated video duration increases, *MiraData* 's performance in motion intensity and aesthetic scores improves, benefiting from the longer video clips in our dataset.

Table 4: **Ablaion on Video Duration.** ↑ and ↓ means higher/lower is better. 1) - 14) are metrics of MiraBench. Refer to Tab. 2 for a detailed explanation of annotation.

| Metrics | | Temporal Motion Strength | | Temporal Consistency | | | 3D Consistency | |
|---|---|---|---|---|---|---|---|---|
| | | 1) DD↑ | 2) TS↑ | 3) DTC↑ | 4) CTC↑ | 5) TMS↑ | 6) MAE$_{\downarrow \times 10^{-2}}$ | 7) RMSE$_{\downarrow \times 10^{-1}}$ |
| | 5s | 7.12 | 22.36 | 20.24 | 20.97 | 21.86 | 91.48 | 12.11 |
| WebVid-10M [9] | 10s | 4.82 | 24.99 | 23.23 | 23.63 | 24.62 | 94.62 | 12.53 |
| | 20s | 4.73 | 63.74 | 57.18 | 59.06 | 62.33 | 99.62 | 13.01 |
| | 5s | 15.46 | 49.47 | 43.78 | 45.95 | 47.24 | 85.27 | 11.74 |
| *MiraData* | 10s | 5.23 | 27.06 | 25.22 | 25.67 | 26.55 | 89.44 | 12.08 |
| | 20s | 6.41 | 84.41 | 76.19 | 78.61 | 82.48 | 96.66 | 12.94 |

| Metrics | | Visual Quality | | Text-Video Alignmnet | | | | |
|---|---|---|---|---|---|---|---|---|
| | | 8) AQ$_{\uparrow \times 10^-}$ | 9) IQ↑ | 10) CA↑ | 11) MOA↑ | 12) BA↑ | 13) SA↑ | 14) OA↑ |
| | 5s | 43.11 | 58.58 | 12.35 | 14.32 | 11.90 | 12.32 | 15.31 |
| WebVid-10M [9] | 10s | 40.98 | 59.60 | 0.12 | 12.99 | 11.61 | 11.91 | 13.65 |
| | 20s | 37.93 | 59.11 | 12.07 | 12.32 | 11.92 | 11.48 | 13.31 |
| | 5s | 49.90 | 63.71 | 12.66 | 14.67 | 12.18 | 12.59 | 16.66 |
| *MiraData* | 10s | 42.60 | 61.47 | 11.97 | 13.62 | 11.17 | 11.77 | 14.94 |
| | 20s | 40.36 | 59.32 | 12.00 | 13.96 | 11.09 | 11.69 | 14.56 |

# C  Limitations and Potential Negative Societal Impacts

## C.1  Limitations and Future Work

Despite the advancements and contributions of our work, several limitations need to be acknowledged and addressed in future research:

- **Dataset Diversity and Coverage.** Although *MiraData* presents a substantial improvement over existing datasets, it may still lack comprehensive diversity in terms of content, genre,

and cultural representation. The dataset's reliance on manually selected channels might introduce a bias towards certain types of videos, potentially affecting the generalizability of models trained on it. Future work could focus on expanding the dataset by including a wider variety of sources and a more balanced representation of different content types.

- **Scalability of the Data Curation Pipeline.** The current data curation pipeline, while effective, might face challenges in scalability, particularly in handling the growing volume of video content and the complexity of annotations required for maintaining high quality. Automating more aspects of the data curation process with more efficient machine learning models could improve scalability.

- **Caption Quality.** While the structured captions in *MiraData* are more detailed than previous datasets, there might still be instances where the captions do not fully capture the nuanced details of the video content. Additionally, using automated captioning tools, despite their high accuracy, can occasionally result in errors or ambiguities. Enhancing the captioning process by integrating human-in-the-loop methods can improve the quality and accuracy of captions. Furthermore, iterative refinement of captions based on feedback from domain experts and end-users could help in generating more precise and informative descriptions.

- **Evaluation Metrics.** The proposed *MiraBench* benchmark, although comprehensive, may not fully cover all aspects of video generation quality, especially those related to subjective human perceptions such as creative quality. Incorporating human evaluations alongside automated metrics can provide a more holistic assessment of generated videos.

## C.2 Potential Negative Societal Impacts and Solutions

The construction of video datasets can lead to possible negative societal impacts such as: (1) Misinformation and Deepfakes. Advances in text-to-video generation models like Sora, particularly those that produce highly realistic and detailed videos, raise significant concerns about the potential for creating deepfakes. These realistic fake videos can be used to spread misinformation, manipulate public opinion, or damage reputations. To solve this, we need to implement robust detection mechanisms and watermarking-generated content to help identify and prevent the misuse of AI-generated videos. Additionally, establishing ethical guidelines and legal frameworks to regulate the use of such technology is crucial. (2) Including Personally Identifiable Information. Collecting videos from various platforms could result in the inclusion of content that contains personally identifiable information, such as faces, locations, or other identifiable features, without consent. We should implement stringent data anonymization techniques and manual review processes to ensure that any PII is either removed or consent is obtained before including such data in the dataset. (3) Bias and Stereotyping. If the dataset used for training contains biased or stereotypical representations, the generated content may perpetuate these biases, leading to harmful societal stereotypes and reinforcing negative perceptions. So, we need to actively curate a diverse and balanced dataset that represents various demographics and perspectives, which can help mitigate bias. Regularly auditing the models for biased outputs and retraining them on more balanced datasets can further reduce this risk.

## D Data Acquisition and License

**Data Acquisition.** Data downloading link is: `https://github.com/mira-space/MiraData`.

**License.** This dataset is made available for informational purposes only. No license, whether implied or otherwise, is granted in or to such dataset (including any rights to copy, modify, publish, distribute and/or commercialize such dataset), unless you have entered into a separate agreement for such rights. Such dataset is provided as-is, without warranty of any kind, express or implied, including any warranties of merchantability, title, fitness for a particular purpose, non-infringement, or that such dataset is free of defects, errors or viruses. In no event will our institution be liable for any damages or losses of any kind arising from the dataset or your use thereof.