# OpenReview forum: "MiraData: A Large-Scale Video Dataset with Long Durations and Structured Captions"
_NeurIPS.cc/2024/Datasets_and_Benchmarks_Track — NeurIPS 2024 Track Datasets and Benchmarks Poster_

### Official Review · Reviewer_k17F · 2024-07-20
**Great paper introducing a long video dataset**

**Rating:** 9
**Confidence:** 5
**Clarity:** The paper is very well written.

**Review:**

**Significance**

This paper approaches a problem of great significance for the video generation community: that of generating long videos. The paper does quality work on this task, producing a good dataset, benchmark, and high quality results in a generation model itself as well.

**Quality**

The data, benchmark, and model are all done correctly, and the results are significantly better than the prior art it's compared to.

Video processing at this scale is very challenging, and the authors handle this task correctly.

**Originality**

The dataset is a new contribution in a challenging field, and multiple aspects of how it's built are interesting and/or novel (filtering process, video grid input, captioning, etc.) when considering that they have to be implemented to work at scale.

The metrics are for the most part already known (except maybe Tracking Strength for example) but the benchmark as a whole that combines all of them is novel.

The model has original contributions in its architecture, which allow it to produce better results than all open video models.

**Clarity**

The paper is very easy to ready and very well written.

**Strengths:**

* The paper is very relevant to the broader research community: generative models are one of the top research areas of the time, and long video generation is one of the most challenging problems there
* The dataset is a significant contribution to the field, as it allows training high quality video models
* The model achieves very good metrics, much better than those of the open models it's compared to
* Experiments and benchmarks are interesting and mostly complete
* Ethics analysis is complete and correct

**Additional Feedback:**

Very exciting work!

**Correctness:**

Dataset construction is solid, benchmark metrics and design are appropriate, and the model construction is solid as well.

**Documentation:**

For the most part, sufficient detail on the dataset is provided. The one thing I find missing is a means to request removal from the dataset, for content owners (as mentioned above)

**Ethics:**

Ethics discussion is complete.

My only concern here is that this is a video dataset sourced from public videos on the internet, which may have copyright concerns that go beyond my knowledge; for this reason I flag for ethics review.

**Limitations:**

The authors have adequately addressed limitations and ethics issues

**Opportunities For Improvement:**

This is a great paper! Still, there are some opportunities for improvement:

* It is mentioned that "*We should implement stringent data anonymization techniques and manual review processes to ensure that any PII is either removed or consent is obtained before including such data in the dataset*"; this seems important enough that it's a good idea to already provide some means to do so. For example the authors could provide a way to allow content owners to request removal from the dataset
* It would be interesting to compare MiraDit generated videos with those of closed models as well. E.g. generating some videos for matching prompts in models like Pika and Gen-2, or using already published videos of models like Sora. This would be an interesting extension to tables like table 2 in the appendix
  * Human evaluation against those closed models would be interesting too
* Ethics discussion could mention the topic of data copyright / data ownership
* Given that Tracking Strength is an evolution over existing metrics, it may benefit from some analysis of how well it aligns with human perception for the same video attribute
* [minor] Tables 2 and 3 in the appendix do not have the best values in bold

**Relation To Prior Work:**

The paper clearly discusses relationship to prior art.

**Summary And Contributions:**

The main contributions of this paper are three:

* A large-scale dataset of long-videos with captions, with the corresponding process for how it's built. This dataset includes much longer videos than other datasets
* A benchmark for evaluating videos, either generated or not, with a wide variety of metrics including aspects of videos that are often overlooked
* A model architecture that can generate long videos, that is trained on both the new dataset and in a previously existing one

The paper also includes:

* Complete analysis of the dataset and the model, against prior art on data and video generation models
* Complete discussion on the ethics aspects of this work

---

> ### Author Response · Authors · 2024-08-17
> **Response to reviewer k17F - 1**
>
> We sincerely appreciate the thoughtful and constructive feedback from the reviewer. We are grateful for the positive assessment of our contributions, including the large-scale long-video dataset, and the benchmarking framework. Below, we  have outlined the reviewer’s questions and provided thorough responses:
>
> > Question 1: Data Anonymization, Consent for PII Removal, and Data Copyright/Ownership in Ethics Discussion
> >
>
> **Answer:**
>
> We fully acknowledge the importance of data anonymization, consent mechanisms for PII removal, and addressing data copyright and ownership. Our approach to data selection and usage reflects these concerns, and we have ensured that all videos sourced from YouTube, Videvo, Pixabay, and Pexels are appropriate for academic use. Specifically, we have carefully chosen YouTube channels that do not impose strict copyright restrictions against academic usage. For videos from Videvo, Pixabay, and Pexels, we have verified that they are suitable for both academic and commercial purposes.
>
> In response to the reviewer’s suggestion, we will provide a clear guideline on our website and dataset distribution points stating:
>
> *"We will remove video samples from our dataset, GitHub repository, or project webpage upon request. Please contact mira-x at googlegroups dot com for removal requests."*
>
> This ensures content owners have a straightforward and transparent way to request the removal of their data. This process will be incorporated into future releases of the dataset.
>
> Moreover, we agree with the importance of discussing data copyright and ownership in more detail. We have already noted on our GitHub page that *"The copyright remains with the original owners of the video."* We will expand the ethics section of the paper to provide a more comprehensive discussion on data copyright, ownership, and transparency in data sourcing. Additionally, we will further emphasize the mechanism by which content creators can request the removal of their content from our dataset, ensuring ethical integrity in data usage.

---

> ### Author Response · Authors · 2024-08-17
> **Response to reviewer k17F - 2**
>
> > Question 2: Human Evaluation with State-of-the-Art Models
> >
>
> **Answer:**
>
> We apologize for the missing of human evaluation in main paper. To provide a more comprehensive assessment, we present the human evaluation results below. We enlisted 6 volunteers to evaluate the entire validation set of MiraBench. Each volunteer was provided with a set of 4 videos generated using OpenSora[1], VideoCrafter2[2], MiraDiT trained on WebVid-10M[3], and MiraDiT trained on MiraData, corresponding to Table 3 in the main paper. The evaluators were asked to rank the four videos from best to worst based on five criteria: (1) motion strength, from strongest to weakest (1-4), (2) temporal consistency, from best to worst (1-4), (3) 3D consistency, from best to worst (1-4), (4) visual quality, from best to worst (1-4), and (5) text-video alignment, from best to worst (1-4). The results of these evaluations are presented in Table 1.
>
> | Metrics | Temporal Motion Strength ↓ | Temporal Consistency ↓ | 3D Consistency ↓ | Visual Quality ↓ | Text-Video Alignment ↓ |
> | --- | --- | --- | --- | --- | --- |
> | OpenSora[1] | 2.6 | 2.5 | 2.6 | 2.8 | 2.9 |
> | VideoCrafter2[2] | 2.9 | **1.8** | 2.3 | **1.4** | 2.3 |
> | MiraDiT (WebVid-10M)[3] | 3.2 | 3.8 | 3.0 | 3.5 | 2.7 |
> | MiraDiT (MiraData) | **1.3** | 1.9 | **2.1** | 2.3 | **2.1** |
>
> Table 1. Human evaluation results of MiraDiT trained on MiraData and WebVid-10M, as well as state-of-the-art open-source methods, OpenSora [1] (DiT-based) and VideoCrafter2 [2] (U-Netbased). Best results are in **bold**.
>
> The results indicate that the human evaluation outcomes are generally aligned with the automatic evaluation metrics in MiraBench, thereby validating the effectiveness of MiraBench. Furthermore, MiraDiT trained on MiraData achieves the highest scores in motion strength, 3D consistency, and text-video alignment, along with comparable performance in temporal consistency and visual quality. This underscores the quality of MiraData. We thank the reviewer for the insightful suggestion and will include human evaluations in our paper’s final version.
>
> [1] Z. Zangwei, P. Xiangyu, L. Shenggui, L. Hongxing, Z. Yukun, L. Tianyi, P. Xiangyu, Z. Zangwei, S. Chenhui, Y. Tom, W. Junjie, and Y. Chenfeng, “Opensora,” 2024.
>
> [2] H. Chen, Y. Zhang, X. Cun, M. Xia, X. Wang, C. Weng, and Y. Shan, “Videocrafter2: Overcoming data limitations for high-quality video diffusion models,” 2024.
>
> [3] M. Bain, A. Nagrani, G. Varol, and A. Zisserman, “Frozen in time: A joint video and image encoder for end-to-end retrieval,” in Proceedings of the IEEE/CVF International Conference on Computer Vision, pp. 1728–1738, 2021.

---

> > ### Comment · Reviewer_k17F · 2024-08-22
> >
> > Thank you for the detailed notes and additional evaluations. I have updated my rating, especially to reflect the additional validation of effectiveness of MiraBench based on the human evaluation. Thank you

---

> > > ### Author Response · Authors · 2024-08-22
> > > **Thanks to Reviewer k17F**
> > >
> > > We sincerely thank the reviewer for the prompt and insightful feedback, as well as for acknowledging the value of our work. The comments and suggestions are valuable in helping us improve our manuscript. We will add the discussion in the rebuttal into the final version of our paper.

---

> ### Author Response · Authors · 2024-08-17
> **Response to reviewer k17F - 3**
>
> > Question 3: Tracking Strength Metric and Human Perception Alignment
> >
>
> **Answer:**
>
> Thank you for the valuable feedback regarding the Tracking Strength metric. As indicated in our human evaluation results, our method, MiraDiT (MiraData), was preferred by more annotators compared to other methods in terms of motion intensity. This aligns with the results presented in Table 3 of the paper, which demonstrates that our approach achieves the best Tracking Strength. Additionally, the ranking of results for open Sora and Video Crafter models corresponds with the order seen in human evaluations, further validating the reliability of the Tracking Strength metric. Notably, MiraDiT (WebVid-10M) ranked lower in terms of motion intensity and other metrics during human evaluation, as some reviewers attributed this to its lower video quality, which also aligns with our findings in Table 3, indicating it had the poorest video quality.
>
> > Question 4: Minor Formatting Issues (Tables 2 and 3 in the Appendix)
> >
>
> **Answer:**
>
> We will address the formatting issues in Tables 2 and 3 to ensure that the best values are clearly highlighted in bold. This will improve the clarity and readability of the results presented in the appendix.
>
> Once again, we extend our sincere thanks your valuable insights and feedback. We will incorporate substantial revisions to our manuscript based on above valuable suggestions, and we will pay special attention to addressing the ethical concerns thoroughly.

---

### Official Review · Reviewer_CiHh · 2024-07-20
**Review of Submission73**

**Rating:** 7
**Confidence:** 4
**Clarity:** The paper is well-written and clear.

**Review:**

Please refer to "Summary And Contributions", "Strengths" and "Opportunities For Improvement".

**Strengths:**

* MiraData distinguishes itself from contemporary T2V training datasets through its extended video durations and structured detail captions.
* The video collection process is meticulously designed, which results in 1) high visual quality, 2) long duration and 3) large motion strength.
* MiraBench presents a comprehensive suite of evaluation criteria. It also introduces some innovative and reasonable designs in the metrics, such as using point tracking to assess video motion strength.
* The empirical results clearly demonstrate the effectiveness of MiraData, highlighting the necessity of utilizing structured and detailed captions in training T2V models.

**Additional Feedback:**

N/A

**Correctness:**

Overall, the methodologies employed for data aggregation and assessment protocols are reasonable. However, there are still several aspects, regarding the data quality and evaluation accuracy, that can be improved. Please refer to “Opportunities for Improvement” for the details.

**Documentation:**

The details on data collection and organization are sufficient and clear.

**Limitations:**

The paper has discussed its limitations in terms of the potential errors in MiraData annotation and inaccuracy of the automatic evaluation metrics. Please refer to “Opportunities for Improvement” for other potential limitations.

**Opportunities For Improvement:**

* A comparison between the automatic metrics in MiraBench and human judgement is missing, making the evaluation results less convincing.
* Two aspects of the data quality are concerning.
  * The current manuscript lacks an assessment of the quality of captions in MiraData, e.g., to what extent do the captions contain inaccurate descriptions of the video content?
  * The stitching operation's performance is not evaluated either.
* The evaluation of T2V models is only conducted on the propped MiraBench. It would also be better to report results on existing T2V benchmarks.
* Some related works on video generation benchmarks [1,2] are not mentioned or discussed in the paper.

I'll consider raising the rating based on the authors' response to the above concerns.

[1] FETV: A Benchmark for Fine-Grained Evaluation of Open-Domain Text-to-Video Generation.

[2] AIGCBench: Comprehensive Evaluation of Image-to-Video Content Generated by AI.

**Relation To Prior Work:**

Some related works on video generation benchmark [1,2] are not mentioned or discussed in the paper.

[1] FETV: A Benchmark for Fine-Grained Evaluation of Open-Domain Text-to-Video Generation.

[2] AIGCBench: Comprehensive Evaluation of Image-to-Video Content Generated by AI.

**Summary And Contributions:**

The contribution of this work is three-fold.
* **MiraData**: MiraData is a novel dataset for training text-to-video (T2V) generation models. Compared with previous text-video datasets, MiraData features longer video duration and structured captions. The videos of the dataset are collected from diverse sources and undergoes meticulous post-processing, which results in 1) high visual quality, 2) long duration and 3) large motion strength. The captions are generated by Panda-70M and GPT-4V, including a short caption, a dense caption and a novelly designed structured caption.
* **MiraBench**: MiraBench consists of 150 text prompts and a suite of metrics evaluating 6 perspectives of the generated videos (e.g., temporal consistency, visual quality and text-video alignment).
* **MiraDiT**: MiraDiT is a T2V model based on Diffusion Transformer. It supports generating videos with dynamic durations and resolutions.

The empirical studies compare MiraDiT trained with different datasets and two SOTA open-sourced T2V models. The results underscore the efficacy of MiraData in enhancing T2V generation across multiple dimensions.

---

> ### Author Response · Authors · 2024-08-17
> **Response to reviewer CiHh - 1**
>
> We sincerely appreciate the reviewer’s insightful comments and recognition of our work. Below, we have outlined the reviewer’s questions and provided thorough responses. We will incorporate additional discussions and results in the revised version of the manuscript.
>
> > Question 1: The results of human judgement
> >
>
> **Answer:**
>
> We apologize for the missing of human evaluation in main paper. To provide a more comprehensive assessment, we present the human evaluation results below. We enlisted 6 volunteers to evaluate the entire validation set of MiraBench. Each volunteer was provided with a set of 4 videos generated using OpenSora[1], VideoCrafter2[2], MiraDiT trained on WebVid-10M[3], and MiraDiT trained on MiraData, corresponding to Table 3 in the main paper. The evaluators were asked to rank the four videos from best to worst based on five criteria: (1) motion strength, from strongest to weakest (1-4), (2) temporal consistency, from best to worst (1-4), (3) 3D consistency, from best to worst (1-4), (4) visual quality, from best to worst (1-4), and (5) text-video alignment, from best to worst (1-4). The results of these evaluations are presented in Table 1.
>
> | Metrics | Temporal Motion Strength ↓ | Temporal Consistency ↓ | 3D Consistency ↓ | Visual Quality ↓ | Text-Video Alignment ↓ |
> | --- | --- | --- | --- | --- | --- |
> | OpenSora 1.1[1] | 2.6 | 2.5 | 2.6 | 2.8 | 2.9 |
> | VideoCrafter2[2] | 2.9 | **1.8** | 2.3 | **1.4** | 2.3 |
> | MiraDiT (WebVid-10M)[3] | 3.2 | 3.8 | 3.0 | 3.5 | 2.7 |
> | MiraDiT (MiraData) | **1.3** | 1.9 | **2.1** | 2.3 | **2.1** |
>
> Table 1. Human evaluation results of MiraDiT trained on MiraData and WebVid-10M, as well as state-of-the-art open-source methods, OpenSora [1] (DiT-based) and VideoCrafter2 [2] (U-Netbased). Best results are in **bold**.
>
> The results indicate that the human evaluation outcomes are generally aligned with the automatic evaluation metrics in MiraBench, thereby validating the effectiveness of MiraBench. Furthermore, MiraDiT trained on MiraData achieves the highest scores in motion strength, 3D consistency, and text-video alignment, along with comparable performance in temporal consistency and visual quality. This underscores the quality of MiraData. We thank the reviewer for the insightful suggestion and will include human evaluations in our paper’s final version.
>
> [1] Z. Zangwei, P. Xiangyu, L. Shenggui, L. Hongxing, Z. Yukun, L. Tianyi, P. Xiangyu, Z. Zangwei, S. Chenhui, Y. Tom, W. Junjie, and Y. Chenfeng, “Opensora,” 2024.
>
> [2] H. Chen, Y. Zhang, X. Cun, M. Xia, X. Wang, C. Weng, and Y. Shan, “Videocrafter2: Overcoming data limitations for high-quality video diffusion models,” 2024.
>
> [3] M. Bain, A. Nagrani, G. Varol, and A. Zisserman, “Frozen in time: A joint video and image encoder for end-to-end retrieval,” in Proceedings of the IEEE/CVF International Conference on Computer Vision, pp. 1728–1738, 2021.

---

> > ### Comment · Reviewer_CiHh · 2024-08-18
> > **Response to Author Rebuttal**
> >
> > Thank you for your comprehensive reply and for addressing my initial concerns regarding data quality, performance on other T2V benchmarks and discussion of related works.
> >
> > I was also pleased to observe the thorough and rigorous human evaluation added in the rebuttal. However, I noted some discrepancies between the results of automatic evaluations and the human judgements. A notable instance is the performance of VideoCrafter2, which, according to automatic metrics, was ranked lowest in terms of **Temporal Consistency**, yet it received the highest ranking under human evaluation criteria.
> >
> > I understand the inherent challenges in developing automatic T2V evaluation metrics that accurately reflect human preference. Nevertheless, I believe it would be better to openly acknowledge these challenges and provide a direct comparison of model rankings using automatic and human evaluation, which may help guide future improvements in T2V metrics. Simply claiming that "the human evaluation outcomes are generally aligned with the automatic evaluation metrics" might mislead.

---

> > > ### Author Response · Authors · 2024-08-18
> > > **Response to Reviewer CiHh's Further Response - 1**
> > >
> > > We thank the reviewer for the prompt reply and the valuable suggestions for improvement. We apologize for the misleading expression regarding the human evaluation metrics. We acknowledge that there are discrepancies between the results of the automatic evaluation and human judgments. Table 3 presents the ranking of each method for both automatic evaluation and human judgments, with the ranks displayed in an "automatic evaluation rank / human judgment rank" format. Since most of the automatic evaluation aspects (Temporal Motion Strength, Temporal Consistency, 3D Consistency, Visual Quality) include multiple metrics for each evaluation dimension, we normalized the scores and summed the normalized scores to determine the final rank (e.g., for OpenSora, the temporal motion strength is 7.65/15.46 + 16.07/49.47 = 0.82). For the text-video alignment, we use Overall Alignment score for automatic evaluation metric ranking.
> > >
> > > | Metrics | Temporal Motion Strength ↓ | Temporal Consistency ↓ | 3D Consistency ↓ | Visual Quality ↓ | Text-Video Alignment ↓ |
> > > | --- | --- | --- | --- | --- | --- |
> > > | OpenSora | 3/2 | 3/3 | 1/3 | 3/3 | 4/4 |
> > > | VideoCrafter2 | 4/3 | 4/1 | 4/2 | 1/1 | 1/2 |
> > > | MiraDiT (WebVid-10M) | 2/4 | 2/4 | 3/4 | 4/4 | 3/3 |
> > > | MiraDiT (MiraData) | 1/1 | 1/2 | 2/1 | 2/2 | 2/1 |
> > >
> > > Table 3. Rank of automatic evaluation and human judgment for OpenSora, VideoCrafter2, MiraDiT trained on WebVid-10M, and MiraDiT trained on MiraData. 1 for best and 4 for worst.
> > >
> > > We acknowledge that the phrase "the human evaluation outcomes are generally aligned with the automatic evaluation metrics" is misleading. What we intended to convey is that the human evaluation results support the conclusions presented in the main paper: MiraDiT trained on MiraData shows significant improvements compared to the WebVid-10M model, while maintaining comparable results compared to state-of-the-art open-source video generation methods such as OpenSora and VideoCrafter2. Moreover, MiraDiT trained on MiraData shows a much better motion strength compared to all other methods.
> > >
> > > Although there are some discrepancies between human evaluation results and automatic evaluation results, they are reasonable:
> > >
> > > - For the **Temporal Consistency** metric in the automatic evaluation, we multiply Tracking Strength by the feature similarities among adjacent video frames. This approach ensures that the metric does not unfairly favor static videos, which would naturally achieve the highest temporal consistency due to their lack of motion (all video frames have a similar feature). By incorporating Tracking Strength, we account for the dynamic nature of the video, ensuring a more reasonable assessment. However, in human evaluations, it is challenging to have annotators consider both metrics simultaneously. Therefore, we simply asked the question, "Is this video temporally consistent?" This approach led to methods like VideoCrafter2 receiving high human evaluation scores, as the videos generated by VideoCrafter2 exhibit very low motion strength.
> > > - For **Temporal Motion Strength** metric, OpenSora, VideoCrafter2, and MiraDiT trained on WebVid-10M, exhibited similarly poor dynamism. This made it difficult for human annotators to determine the relative ranking of these three methods in terms of motion strength. As a result, the human evaluation results show similar Temporal Motion Strength scores among these methods. Additionally, due to time constraints, we were only able to enlist 6 volunteers for the human judgment task, which introduced some variability in cases where the dynamism was similarly weak. We had the opportunity to ask one of the annotators why he ranked MiraDiT trained on WebVid-10M last in motion strength for one set of videos. The response was, “The videos generated by OpenSora, VideoCrafter2, and MiraDiT trained on WebVid-10M seem similar in motion strength, but the aesthetics of MiraDiT trained on WebVid-10M are not appealing.” We will include more annotators for human evaluation and restrict the annotation criterion in the final version of the paper to minimize the impact of this kind of randomness.
> > > - For **3D consistency** metric, we find it really hard for human beings to accurately judge whether a video's scene is 3D consistency (e.g., alignment with 3D modeling standards and physical optics projection). However, automatic metrics also face difficulties in determining 3D consistency due to unignorable calculation errors in 3D modeling methods. Therefore, we believe that the most effective approach is to incorporate both automated and human indicators in the evaluation process.

---

> > > ### Author Response · Authors · 2024-08-18
> > > **Response to Reviewer CiHh's Further Response - 2**
> > >
> > > We sincerely apologize once again for the misleading expression. We will openly acknowledge these challenges and include both automatic and human evaluation results, along with our thorough discussion on this issue, in the final version of our paper. We are grateful to the reviewer for highlighting this concern and for initiating a discussion on the challenges involved in developing automatic T2V evaluation metrics. This discussion will significantly improve the quality of our paper and prompted us to reflect on the evaluation system. In the final version, we will include more annotators for the evaluation, ensuring that the human evaluation results are more robust and valuable to the research community.

---

> > > > ### Comment · Reviewer_CiHh · 2024-08-18
> > > > **Reply to Author's Further Response**
> > > >
> > > > I thank the authors for the follow-up response. I appreciate the enhanced discussion on automatic and human evaluation, which offers valuable insights. The author rebuttal has effectively addressed most of my concerns, revealing the value of this work beyond its limitations. Consequently, I decide to adjust my rating from 5 to 7.
> > > >
> > > > Regarding the **Temporal Consistency** metric, I would like to offer an additional comment: Given that temporal consistency and temporal motion strength are already reported as distinct results, I question the necessity to multiply *Tracking Strength* in the calculation of temporal consistency values.

---

> > > > > ### Author Response · Authors · 2024-08-19
> > > > > **Reply to the Reviewer CiHh's Further Response**
> > > > >
> > > > > We would like to express our sincere gratitude to the reviewer for their recognition and valuable suggestions regarding our work. To avoid temporal consistency negative correlated with motion strength (a video with higher motion strength tends to get a lower score in temporal consistency), we have multiplied the tracking strength in the calculation of temporal consistency metrics.
> > > > >
> > > > > Our intention was to provide a more reasonable understanding of this concept. However, we acknowledge that temporal consistency and temporal motion strength have already been reported as distinct metrics. In response to this, we will add a column in our tables reporting the original temporal consistency score (without multiplying tracking strength). We believe this adjustment will enhance the audience's understanding of MiraBench. We are thankful to the reviewer for all these insightful suggestions, and will consider developing new metrics for temporal consistency that inherently take motion strength into consideration.

---

> ### Author Response · Authors · 2024-08-17
> **Response to reviewer CiHh - 2**
>
> > Question 2: The assessment of data processing quality.
> >
>
> **Answer:**
>
> We thank the reviewer for pointing out the missing of quality assessment for data processing.  In response, we have conducted an evaluation of the quality of captioning and stitching operations across each data category through human evaluation.
>
> Specifically, 6 volunteers participated in this evaluation, with each individual given 2 randomly sampled videos from each category along with their corresponding captions. Our video dataset comprises 8 categories from YouTube videos (7 manually selected categories and HD-VILA-100M), as well as videos from Videvo, Pixabay, and Pexels, resulting in a total of 11 categories. The volunteers were asked to evaluate the following aspects:
>
> 1. **Video Splitting and Stitching:** Is the video split and stitched correctly? This refers to clips without shot transitions or with strong continuity between transitions.
> 2. **Short Caption Accuracy:** Does the short caption accurately describe the video?
> 3. **Dense Caption Accuracy:** Does the dense caption accurately and comprehensively describe the video?
> 4. **Main Object Caption Accuracy:** Does the main object caption accurately and comprehensively describe the primary object in the video?
> 5. **Background Caption Accuracy:** Does the background caption accurately and comprehensively describe the video's background?
> 6. **Camera Caption Accuracy:** Does the camera caption accurately and comprehensively describe the camera movements in the video?
> 7. **Style Caption Accuracy:** Does the style caption accurately and comprehensively describe the style of the video?
>
> The results are shown in Table 2.
>
> |  | video stitch | short caption | dense caption | main object | background | camera | style |
> | --- | --- | --- | --- | --- | --- | --- | --- |
> | **Average Score** | 0.84 | 0.96 | 0.97 | 0.94 | 0.93 | 0.92 | 0.98 |
>
> Table 2. Human evaluation of video stitching and captioning.
>
> The results demonstrate that both video stitching and captioning exhibit a relatively high level of accuracy, making them suitable for model training. Moving forward, we will explore additional methods to further evaluate the accuracy of stitching and captioning, as well as work on developing improved techniques for these processes.

---

> ### Author Response · Authors · 2024-08-17
> **Response to reviewer CiHh - 3**
>
> > Question 3: Results on existing T2V Benchmark
> >
>
> **Answer:**
>
> To assess MiraDiT's performance on other benchmark datasets, we test the performance of MiraDiT on the recent text-to-video benchmark, T2V-CompBench[4]. T2V-CompBench includes 7 metrics designed to evaluate the alignment of generated videos with the corresponding text prompts:
>
> 1. **Consistent Attribute Binding**: Evaluates whether object attributes remain consistent throughout the generated video frames.
> 2. **Dynamic Attribute Binding**: Assesses if the generated video accurately reflects changes in object attributes.
> 3. **Spatial Relationship**: Determines if the generated video adheres to the spatial relationships specified in the text prompt.
> 4. **Motion Binding**: Assesses the correctness of the object's motion direction in the generated video.
> 5. **Action Binding**: Evaluates the accuracy of the object action categories in the generated video.
> 6. **Object Interactions**: Tests the model's ability to generate dynamic interactions between objects.
> 7. **Generative Numeracy**: Evaluates the accuracy in the number of objects generated as specified in the text prompt.
>
> The results of MiraDiT trained on MiraData and WebVid-10M[5] compared to previous open-source methods is shown in Table 3.
>
> | Method | 1. Consist-attr ↑ | 2.Dynamic-attr ↑ | 3. Spatial ↑ | 4. Motion ↑ | 5. Action ↑ | 6. Interaction ↑ | 7. Numeracy ↑ |
> | --- | --- | --- | --- | --- | --- | --- | --- |
> | ModelScope  | 0.5483 | 0.1654 | 0.4220 | 0.2552 | 0.4880 |  0.7075 | 0.2066 |
> | ZeroScope | 0.4495 | 0.1086 | 0.4073 | 0.2319 |  0.4620 | 0.5550 | 0.2378 |
> | Latte | 0.5325 | 0.1598 | 0.4476 | 0.2187 | 0.5200 | 0.6625 | 0.2187 |
> | Show-1 | 0.6388 | 0.1828 | 0.4649 | 0.2316 | 0.4940 | **0.7700** | 0.1644 |
> | VideoCrafter2 |  0.6750 | 0.1850 | 0.4891 | 0.2233 | 0.5800 | 0.7600 | 0.2041 |
> | Open-Sora 1.1 | 0.6370 | 0.1762 | **0.5671** | 0.2317 | 0.5480 | 0.7625 | 0.2363 |
> | Open-Sora 1.2 | 0.6600 | 0.1714 | 0.5406 | 0.2388 | 0.5717 | 0.7400 | 0.2556 |
> | Open-Sora-Plan v1.0.0 | 0.5088 | 0.1562 | 0.4481 | 0.2147 | 0.5120 | 0.6275 | 0.1650 |
> | Open-Sora-Plan v1.1.0 | **0.7413** | 0.1770 | 0.5587 | 0.2187 | **0.6780** | 0.7275 | **0.2928** |
> | AnimateDiff | 0.4883 | 0.1764 | 0.3883 | 0.2236 | 0.4140 | 0.6550 | 0.0884 |
> | VideoTetris | 0.7125 | 0.2066 | 0.5148 | 0.2204 | 0.5280 | 0.7600 | 0.2609 |
> | LVD | 0.5595 | 0.1499 | 0.5469 | **0.2699** | 0.4960 | 0.6100 | 0.0991 |
> | MagicTime | - | 0.1834 | - | - | - | - | - |
> | MiraDiT (WebVid-10M) | 0.6012 | 0.1972 | 0.4438 | 0.2250 | 0.5156 | 0.6075 | 0.1909 |
> | MiraDiT (MiraData) | 0.6825 | **0.2302** | 0.4622 | 0.2321 | 0.6340 | 0.7373 | 0.2234 |
> | rank of MiraDiT (MiraData) | 3/15 | 1/15 | 8/15 | 4/15 | 2/15 | 6/15 | 6/15 |
>
> Table 3. T2V-CompBench evaluation results of MiraDiT trained on MiraData and WebVid-10M. Metrics 1-7 are consistent with the description of metrics above. Best results are shown in **bold**. We also list the rank of MiraDiT trained on MiraData at the last row.
>
> Results show that MiraDiT trained on MiraData achieves much better results on all metrics compare to that trained on WebVid-10M. Moreover, MiraDiT trained on MiraData have the best results on Dynamic Attribute Binding, further illustrates the advantages of training with high-dynamic, detailed-captioned data. MiraDiT trained on MiraData also achieves a relatively advanced results in all open-source text-to-video generation models. However, we must point out, that this comparison is unfair, as different models were trained using different computational resources and distinct models, making it impossible to assess the quality of MiraData relative to other training data. Moreover, the evaluation prompts in T2V-CompBench primarily consist of short captions with only a single simple sentence, which limits MiraData’s ability to fully showcase its strengths.
>
> [4] Sun K, Huang K, Liu X, et al. T2V-CompBench: A Comprehensive Benchmark for Compositional Text-to-video Generation[J]. arXiv preprint arXiv:2407.14505, 2024.
>
> [5] M. Bain, A. Nagrani, G. Varol, and A. Zisserman, “Frozen in time: A joint video and image encoder for end-to-end retrieval,” in Proceedings of the IEEE/CVF International Conference on Computer Vision, pp. 1728–1738, 2021.

---

> ### Author Response · Authors · 2024-08-17
> **Response to reviewer CiHh - 4**
>
> > Question 4: Missing discussion of related work
> >
>
> **Answer:**
>
> We thank the reviewer for pointing out missing related work. FETV[6] is a text-to-video benchmark targeted for fine-grained evaluation of 3 different categories of text prompts. It also develops improved CLIPScore and FVD that exhibit higher correlation with human evaluation. AIGCBench[7] is an image-to-video generation benchmark with comprehensive evaluation metrics spanning control-video alignment, motion effects, temporal consistency, and video quality. Similar to FETV, MiraBench also focus on different categories of text prompt and evaluate text-video alignment on 5 aspects: camera, main object, background, style, and overall description. Compared to these two works, MiraBench shows an enhancement and expansion on metrics set, especially in dynamic degree.  We would further discuss these related work in the revised version, which would significantly improved the quality and readability of the paper.
>
> [6] Liu Y, Li L, Ren S, et al. Fetv: A benchmark for fine-grained evaluation of open-domain text-to-video generation[J]. Advances in Neural Information Processing Systems, 2024, 36.
>
> [7] Fan F, Luo C, Gao W, et al. AIGCBench: Comprehensive evaluation of image-to-video content generated by AI[J]. BenchCouncil Transactions on Benchmarks, Standards and Evaluations, 2023, 3(4): 100152.

---

### Official Review · Reviewer_AiEx · 2024-07-23
**MiraData: A Large-Scale Video Dataset with Long Durations and Structured Captions**

**Rating:** 5
**Confidence:** 3
**Clarity:** The paper lacks detailed explanations…

**Review:**

This paper introduces a high quality video dataset, MiraData designed for generating realistic videos. It features long videos, high motion intensity and detailed structured captions. In addition to this dataset, the authors propose a new benchmark aimed at improving the assessment of temporal consistency and motion intensity in video generation tasks with MiraBench. Furthermore, a video generation model, MiraDiT is developed, tailored for specific applications, with pipeline based on diffusion transformer to assess the effectiveness of MiraData.
However, when reviewing the supplemental demo videos, the claims of generating realistic video is doubtful. Videos in demo are very short. Also the word cloud or qualitative comparison with only 8 frame do not seem convincing.
Authors claimed that "To evaluate the effectiveness of MiraData on long-duration video generation, we train a dynamic frame rate video generation model that supports arbitrary length video generation from 0 to 20s on MiraData and WebVid respectively. ... The experimental results demonstrate that our MiraData achieves significantly better motion strength and dynamic degree compared to the model
trained on WebVid-10M," I wonder if the montion strength and dynamic degree can associate the effectiveness of the dataset.
The supplemental materials (demo video) don't show the benefit.
The paper acknowledges several inherent limitations of the proposed dataset and methods.

**Strengths:**

It is important to develop a large-scale text-video dataset for long-duration videos with detailed captions, and a comprehensive evaluation framework for video generation models.
    Diverse sources of videos from multiple platforms, ensuring a broad range of content and high visual quality
    The dataset tries to enhance the ability to generate long videos and high motion intensity
    Detailed annotation process with advanced models and methods for annotation, including structured captions
    Diverse evaluation framework with extensive metrics covering multiple aspects of video generation, including temporal consistency and motion strength

**Additional Feedback:**

n/a

**Correctness:**

Yes, however, 17 evaluation metrics designed are not clearly explained. So, it is difficult to judge the correctness of these evaluation metrics.

**Documentation:**

Authors tried to cover large topics briefly. The page limit is real but maybe the paper should split the evaluation metrics as a separate paper to give proper discussion on 17 evaluation metrics.

**Ethics:**

Minor ethical concerns which is already acknowledged in the paper.

**Limitations:**

Even with advanced captioning methods, there could be variations in how captions depict similar scenes or actions, impacting the accuracy of the model's learning accuracy.
    The reliance on particular video platforms (e.g., YouTube, stock video sites) might introduce platform-specific biases and limit the diversity of video types.
    The enhanced video generation capabilities could inadvertently lead to privacy concerns or misuse in creating misleading or harmful content.

**Opportunities For Improvement:**

Evaluation frame work does not address the semantic quality of the video.
What does "structured caption" means or the value of the structure? It was not well discussed.
Typically models are not good at cross-dataset generalization. The MiraDiT;s performance on other dataset need to be more thoroughly addressed.
Also authors point that The results in Tab. 4 demonstrate that longer and more detailed captions do not necessarily improve the visual quality of the generated videos. However, they offer significant benefits in terms of increased dynamics, enhanced temporal consistency, more accurate generation control, and better alignment between the text and the generated video content.
It is not clear how visual quality is measured vs. text-video alignment metric, for example.

**Relation To Prior Work:**

covers sufficient prior work

**Summary And Contributions:**

The paper introduces MiraData, a large-scale text-video dataset for long-duration videos with detailed captions, and MiraBench, a comprehensive evaluation framework for video generation models. The MiraDiT model, designed based on the Diffusion Transformer, is presented to validate its effectiveness in generating consistent, high-motion long videos.

The dataset was sourced from YouTube, Videvo, Pixabay, and Pexels, underwent a rigorous five-step collection and annotation process, and featured detailed captions and structured descriptions.

Mira Bench provides a thorough evaluation framework with 17 metrics from six perspectives and emphasizes the evaluation of models using general prompts and 3D consistency. The paper also provides extensive experimental validation and discusses potential limitations and ethical considerations while advancing the field of high-quality, temporally consistent long-video generation.

Their contribution includes dataset which is an improvement over existing datasets ensuring long-duration videos with high motion intensity and detailed annotations and highlighting the effectiveness of detailed and structured captions.

---

> ### Author Response · Authors · 2024-08-17
> **Response to reviewer AiEx -1**
>
> We sincerely thank the reviewer for the insightful comments and recognition of our work. Below, we have summarized the reviewer’s questions and provided detailed responses. We will include new discussions and results in the revised version of the manuscript.
>
> > Question 1: The effectiveness of the datasets
> >
>
> **Answer:**
>
> Previous video datasets primarily consist of short clips with low motion, thus MiraData is introduced to address this gap. As demonstrated in Table 3 of the main paper, MiraDiT, trained on MiraData, not only exhibits best temporal motion strength and consistency but also maintains a comparable visual quality (second best overall). This validates the effectiveness of MiraData.
>
> Due to computational limitations, we admit that MiraDiT does not achieve the same level of quality as state-of-the-art video generation models that utilize thousands of GPUs for training (e.g., [Sora](https://openai.com/index/sora/?ref=botark.com), [Kling](https://kling.kuaishou.com/en)). However, our key contribution is the creation of a high-quality dataset characterized by long duration and significant motion strength. We apologize for providing only short video samples in the supplementary files and offer additional examples for visualization here: https://drive.google.com/drive/folders/1bqbb1e1ijDH68L1imLiK0oEsc_SqGC5z?usp=sharing . Two sets of results are provided in the link: one with a resolution of 384x240 for 20 seconds, and the other with a resolution of 768x480 for 10 seconds.
>
> > Question 2: More detailed explanation of the evaluation metrics. The evaluation of the semantic quality of video.
> >
>
> **Answer:**
>
> We apologize for the lack of clarity in the explanation of our proposed benchmark, particularly in how MiraBench evaluates video semantic quality. We offer a more detailed explanation below:
>
> Our evaluation metrics encompass six key perspectives: temporal consistency, temporal motion strength, 3D consistency, visual quality, text-video alignment, and distribution consistency. Among these, **temporal consistency**, **3D consistency**, and **visual quality** specifically focus on **video semantic quality** in terms of **temporal coherence**, **physical 3D reconstruction**, and **frame-wise quality**, respectively. Additionally, **text-video alignment** evaluates the **semantic alignment between text and video semantics**. Below is a more detailed explanation:
>
> - **Temporal consistency** includes three metrics that assess feature similarity between adjacent frames using DINO[1] and CLIP[2], and evaluate motion prior smoothness using AMT[3]. These metrics determine whether the semantic information between frames remains consistent, thereby identifying any temporal degradation in video semantics.
> - **3D consistency** is measured by calculating the Mean Absolute Error and Root Mean Square Error in 3D reconstruction following GVGC[4]. These metrics evaluate the consistency of physical 3D semantic information across frames.
> - **Visual quality** assesses the semantic quality on a per-frame basis. Aesthetic quality is determined by calculating an aesthetic score for each frame, while imaging quality evaluate potential distortions in video frames (e.g., over-exposure).
> - **Text-video alignment** use ViCLIP[5] to evaluate video and text semantic consistency. Based on our structured text prompt design, we calculate text-video alignment across five types of prompts: camera, main object, background, style, and overall description.
>
> Supplementary files and our source code also provide more detailed information on how evaluation metrics are calculated. We express our gratitude for the reviewer's suggestions, and will include a more detailed explanation as well as separate documnents of the proposed benchmark in the final version of our manuscript.
>
> [1] M. Oquab, T. Darcet, T. Moutakanni, H. Vo, M. Szafraniec, V. Khalidov, P. Fernandez, D. Haziza, F. Massa, A. El-Nouby, et al., “Dinov2: Learning robust visual features without supervision,” arXiv preprint arXiv:2304.07193, 2023.
>
> [2] A. Radford, J. W. Kim, C. Hallacy, A. Ramesh, G. Goh, S. Agarwal, G. Sastry, A. Askell, P. Mishkin, J. Clark, et al., “Learning transferable visual models from natural language supervision,” in International conference on machine learning, pp. 8748–8763, PMLR, 2021.
>
> [3] Z. Li, Z.-L. Zhu, L.-H. Han, Q. Hou, C.-L. Guo, and M.-M. Cheng, “Amt: All-pairs multi-field transforms for efficient frame interpolation,” in Proceedings of the IEEE/CVF Conference on Computer Vision and Pattern Recognition, pp. 9801–9810, 2023.
>
> [4] X. Li, D. Zhou, C. Zhang, S. Wei, Q. Hou, and M.-M. Cheng, “Sora generates videos with stunning geometrical consistency,” arXiv preprint arXiv:2402.17403, 2024.
>
> [5] Y. Wang, Y. He, Y. Li, K. Li, J. Yu, X. Ma, X. Li, G. Chen, X. Chen, Y. Wang, et al., “Internvid: A large-scale video-text dataset for multimodal understanding and generation,” arXiv preprint arXiv:2307.06942, 2023.

---

> ### Author Response · Authors · 2024-08-17
> **Response to reviewer AiEx -2**
>
> > Question 3: What does “structured caption” mean?
> >
>
> **Answer:**
>
> We apologize for the lack of clarity of the term “structured caption.” Each video in MiraData is accompanied by six types of captions: short caption, dense caption, main object caption, background caption, camera caption, and style caption. These captions are interrelated, which is why we refer a combination of them as structured captions.
>
> The short caption provides a summary of the most critical information in a video. The dense caption offers a more comprehensive description, encompassing the main object, background, camera, and style, along with additional details. The main object caption, background caption, camera caption, and style caption each focus on a specific aspect of the video. This creates a hierarchical structure, progressing from a general overview to a more detailed description: {short caption} → {main object caption, background caption, camera caption, and style caption} → {dense caption}. We use structured captions in model training, which is a combination of short caption, main object caption, background caption, camera caption, and style caption, and dense caption.
>
> We appreciate the reviewer’s feedback and will include this explanation in the final version of our paper.

---

> ### Author Response · Authors · 2024-08-17
> **Response to reviewer AiEx -3**
>
> > Question 4: The MiraDiT’s performance on other datasets
> >
>
> **Answer:**
>
> To assess MiraDiT's performance on other benchmark datasets, we test the performance of MiraDiT on the recent text-to-video benchmark, T2V-CompBench[6]. T2V-CompBench includes 7 metrics designed to evaluate the alignment of generated videos with the corresponding text prompts:
>
> 1. **Consistent Attribute Binding**: Evaluates whether object attributes remain consistent throughout the generated video frames.
> 2. **Dynamic Attribute Binding**: Assesses if the generated video accurately reflects changes in object attributes.
> 3. **Spatial Relationship**: Determines if the generated video adheres to the spatial relationships specified in the text prompt.
> 4. **Motion Binding**: Assesses the correctness of the object's motion direction in the generated video.
> 5. **Action Binding**: Evaluates the accuracy of the object action categories in the generated video.
> 6. **Object Interactions**: Tests the model's ability to generate dynamic interactions between objects.
> 7. **Generative Numeracy**: Evaluates the accuracy in the number of objects generated as specified in the text prompt.
>
> The results of MiraDiT trained on MiraData and WebVid-10M[7] compared to previous open-source methods is shown in Table 1.
>
> | Method | 1. Consist-attr ↑ | 2.Dynamic-attr ↑ | 3. Spatial ↑ | 4. Motion ↑ | 5. Action ↑ | 6. Interaction ↑ | 7. Numeracy ↑ |
> | --- | --- | --- | --- | --- | --- | --- | --- |
> | ModelScope  | 0.5483 | 0.1654 | 0.4220 | 0.2552 | 0.4880 |  0.7075 | 0.2066 |
> | ZeroScope | 0.4495 | 0.1086 | 0.4073 | 0.2319 |  0.4620 | 0.5550 | 0.2378 |
> | Latte | 0.5325 | 0.1598 | 0.4476 | 0.2187 | 0.5200 | 0.6625 | 0.2187 |
> | Show-1 | 0.6388 | 0.1828 | 0.4649 | 0.2316 | 0.4940 | **0.7700** | 0.1644 |
> | VideoCrafter2 |  0.6750 | 0.1850 | 0.4891 | 0.2233 | 0.5800 | 0.7600 | 0.2041 |
> | Open-Sora 1.1 | 0.6370 | 0.1762 | **0.5671** | 0.2317 | 0.5480 | 0.7625 | 0.2363 |
> | Open-Sora 1.2 | 0.6600 | 0.1714 | 0.5406 | 0.2388 | 0.5717 | 0.7400 | 0.2556 |
> | Open-Sora-Plan v1.0.0 | 0.5088 | 0.1562 | 0.4481 | 0.2147 | 0.5120 | 0.6275 | 0.1650 |
> | Open-Sora-Plan v1.1.0 | **0.7413** | 0.1770 | 0.5587 | 0.2187 | **0.6780** | 0.7275 | **0.2928** |
> | AnimateDiff | 0.4883 | 0.1764 | 0.3883 | 0.2236 | 0.4140 | 0.6550 | 0.0884 |
> | VideoTetris | 0.7125 | 0.2066 | 0.5148 | 0.2204 | 0.5280 | 0.7600 | 0.2609 |
> | LVD | 0.5595 | 0.1499 | 0.5469 | **0.2699** | 0.4960 | 0.6100 | 0.0991 |
> | MagicTime | - | 0.1834 | - | - | - | - | - |
> | MiraDiT (WebVid-10M) | 0.6012 | 0.1972 | 0.4438 | 0.2250 | 0.5156 | 0.6075 | 0.1909 |
> | MiraDiT (MiraData) | 0.6825 | **0.2302** | 0.4622 | 0.2321 | 0.6340 | 0.7373 | 0.2234 |
> | rank of MiraDiT (MiraData) | 3/15 | 1/15 | 8/15 | 4/15 | 2/15 | 6/15 | 6/15 |
>
> Table 1. T2V-CompBench evaluation results of MiraDiT trained on MiraData and WebVid-10M. Metrics 1-7 are consistent with the description of metrics above. Best results are shown in **bold**. We also list the rank of MiraDiT trained on MiraData at the last row.
>
> Results show that MiraDiT trained on MiraData achieves much better results on all metrics compare to that trained on WebVid-10M. Moreover, MiraDiT trained on MiraData have the best results on Dynamic Attribute Binding, further illustrates the advantages of training with high-dynamic, detailed-captioned data. MiraDiT trained on MiraData also achieves a relatively advanced results in all open-source text-to-video generation models. However, we must point out, that this comparison is unfair, as different models were trained using different computational resources and distinct models, making it impossible to assess the quality of MiraData relative to other training data. Moreover, the evaluation prompts in T2V-CompBench primarily consist of short captions with only a single simple sentence, which limits MiraData’s ability to fully showcase its strengths.
>
> [6] Sun K, Huang K, Liu X, et al. T2V-CompBench: A Comprehensive Benchmark for Compositional Text-to-video Generation[J]. arXiv preprint arXiv:2407.14505, 2024.
>
> [7] M. Bain, A. Nagrani, G. Varol, and A. Zisserman, “Frozen in time: A joint video and image encoder for end-to-end retrieval,” in Proceedings of the IEEE/CVF International Conference on Computer Vision, pp. 1728–1738, 2021.

---

> ### Author Response · Authors · 2024-08-17
> **Response to reviewer AiEx -4**
>
> > Question 5: How visual quality is measured vs. text-video alignment metric
> >
>
> **Answer:**
>
> As demonstrated in Section 4.2, we calculate visual quality using the frame-wise LAION[8]  aesthetic score (AQ) and the MUSIQ[9] score (IQ). For text-video alignment, we calculate ViCLIP[5] scores between the text prompt and the generated video. Table 4 presents metrics for three settings: short caption, dense caption, and structural caption, with the structural caption being a combination of all caption types. The results indicate that while structural and dense captions do not necessarily achieve better visual quality (as measured by AQ and IQ) compared to short captions, they significantly improve overall caption alignment (OA), which measures the ViCLIP score between the generated video and the dense text caption. Additionally, structural and dense captions exhibit notably better temporal motion strength and temporal consistency scores. We will provide more detailed explanation on this in the revised paper version.
>
> [8] R. Rombach, A. Blattmann, D. Lorenz, P. Esser, and B. Ommer, “High-resolution image synthesis with latent diffusion models,” in Proceedings of the IEEE/CVF Conference on Computer Vision and Pattern Recognition (CVPR), pp. 10684–10695, June 2022.
>
> [9] J. Ke, Q. Wang, Y. Wang, P. Milanfar, and F. Yang, “Musiq: Multi-scale image quality transformer,” in Proceedings of the IEEE/CVF international conference on computer vision, pp. 5148–5157, 2021.
>
> > Question 6: Possible limitations in captioning, platform-specific biases, and the ethical implications
> >
>
> **Answer:**
>
> 1. **Captioning Variations:**
> We acknowledge that even with advanced captioning methods, there can be variations in how captions describe similar scenes or actions, which could influence the model's learning accuracy. To mitigate this, we have implemented structured captions to make sure the captioning model can provide a more consistent and comprehensive description across different aspects of the video. We will continue to explore ways to improve this issue in our future endeavors.
> 2. **Platform-Specific Biases:**
> We have made an effort to source videos from a variety of platforms, including YouTube, Videovo, Pixabay, and Pexels, to capture a broad range of content. However, we recognize that platform-specific biases may still exist. Moving forward, we plan to diversify our data sources further and incorporate additional checks to identify and mitigate any biases that might arise from platform-specific content.
> 3. **Ethical Considerations and Privacy Concerns:**
> We fully agree that the enhanced capabilities of video generation models could raise privacy concerns or lead to the misuse of technology in creating misleading or harmful content. We have established guidelines and restrictions to prevent misuse, and we are actively collaborating with the broader research community to address these challenges.
>
> We appreciate your valuable insights and will continue to prioritize these concerns in our ongoing research and development efforts. Thank you once again for your thoughtful review.

---

> ### Author Response · Authors · 2024-08-22
> **The authors are looking forward to your feedback.**
>
> Dear Reviewer AiEx,
>
> We sincerely appreciate the time and effort you have invested in reviewing our manuscript and for providing insightful suggestions to enhance our work. We understand the demands of your schedule and the potential volume of papers you review. In our previous response, we carefully addressed each of your remaining concerns in detail. We look forward to receiving your further feedback on our responses.
>
> Best regards,
>
> The Authors

---

> ### Author Response · Authors · 2024-08-28
> **The rebuttal period has only 3 days remaining. The authors are eagerly awaiting your feedback.**
>
> Dear Reviewer AiEx,
>
> We sincerely appreciate the time and effort you have dedicated to reviewing our manuscript and for providing insightful suggestions to enhance our work. We understand that you may have a tight schedule and are likely reviewing multiple papers. However, as the rebuttal period has only 3 days remaining, we hope to initiate a discussion soon. We look forward to receiving your further feedback on our responses.
>
> Best regards,
>
> The Authors

---

### Author Rebuttal · Authors · 2024-08-17

We sincerely appreciate all the reviewers for their constructive feedback and recognition of our work. We are especially grateful for acknowledging the strengths of our work, including: (1) the significance of the video dataset with **long duration and structured captions**, which distinguishes MiraData apart from previous video datasets (Reviewers AiEx, CiHh, k17F); (2) the **detailed and diligent data collection and annotation process** (Reviewers AiEx, CiHh, k17F); (3) the **diverse data sources with high-quality videos** (Reviewers AiEx, k17F); (4) the **comprehensive evaluation framework** (Reviewers AiEx, CiHh, k17F); and (5) the **convincing experimental results** (Reviewers CiHh, k17F).

We would also like to express our sincere gratitude to the reviewers for their insightful identification of areas where our manuscript could be improved. We have carefully summarized each reviewer’s questions and our responses under each reviewer's review, and we will incorporate substantial revisions to our manuscript based on these valuable suggestions.

---

### Decision · Program_Chairs · 2024-09-26

**Decision:**

Accept (Poster)

**Comment:**

The reviews for this paper are mixed, with two reviewers recommending acceptance and one rating it just below the acceptance threshold. After carefully reading the comments from the less favorable review, it seems that the concerns raised primarily require further clarification and explanation. I believe the authors have sufficiently addressed these points, even though the reviewer has not provided additional feedback. Overall, all three reviewers acknowledge the relevance of the paper and its significant contributions to the field.